**DOI: 10.1038/ncomms15198**　　**OPEN**

# Giant five-photon absorption from multidimensional core-shell halide perovskite colloidal nanocrystals

Weiqiang Chen[1], Saikat Bhaumik[2], Sjoerd A. Veldhuis[2], Guichuan Xing[1,†], Qiang Xu[1], Michael Grätzel[2,3], Subodh Mhaisalkar[2,4], Nripan Mathews[2,4] & Tze Chien Sum[1]

Multiphoton absorption processes enable many technologically important applications, such as *in vivo* imaging, photodynamic therapy and optical limiting, and so on. Specifically, higher-order nonlinear absorption such as five-photon absorption offers significant advantages of greater spatial confinement, increased penetration depth, reduced autofluorescence, enhanced sensitivity and improved resolution over lower orders in bioimaging. Organic chromophores and conventional semiconductor nanocrystals are leaders in two-/three-photon absorption applications, but face considerable challenges from their small five-photon action cross-sections. Herein, we reveal that the family of halide perovskite colloidal nanocrystals transcend these constraints with highly efficient five-photon-excited upconversion fluorescence—unprecedented for semiconductor nanocrystals. Amazingly, their multidimensional type I (both conduction and valence band edges of core lie within bandgap of shell) core–shell (three-dimensional methylammonium lead bromide/two-dimensional octylammonium lead bromide) perovskite nanocrystals exhibit five-photon action cross-sections that are at least 9 orders larger than state-of-the-art specially designed organic molecules. Importantly, this family of halide perovskite nanocrystals may enable fresh approaches for next-generation multiphoton imaging applications.

[1] Division of Physics and Applied Physics, School of Physical and Mathematical Sciences, Nanyang Technological University (NTU), 21 Nanyang Link, SPMS-PAP 03-05, Singapore 637371, Singapore. [2] Energy Research Institute @ NTU (ERI@N), Research Techno Plaza, X-Frontier Block, Level 5, 50 Nanyang Drive, Singapore 637553, Singapore. [3] Laboratory of Photonics and Interfaces, Department of Chemistry and Chemical Engineering, Swiss Federal Institute of Technology, Station 6, Lausanne 1015, Switzerland. [4] School of Materials Science and Engineering, NTU, Nanyang Avenue, Singapore 639798, Singapore. [†] Present address: Institute of Applied Physics and Materials Engineering, Faculty of Science and Technology, University of Macau, Macao SAR 999078, China. Correspondence and requests for materials should be addressed to N.M. (email: Nripan@ntu.edu.sg) or to T.C.S. (email: Tzechien@ntu.edu.sg).

Multiphoton absorption (MPA) involves the simultaneous absorption of multiple monochromatic infrared photons that excites an electron to a higher energy state, whereupon its subsequent relaxation yields the emission of a shorter wavelength photon. Such frequency-upconverted fluorescence possesses many attractive merits over linearly excited fluorescence (for example, high spatial confinement, long penetration depth and low biological damage)[1–6] due to the longer infrared excitation wavelengths. Two-photon/three-photon absorption (2PA/3PA) have found widespread applications in three-dimensional (3D) biomedical imaging[1–6], optical power limiting[7], sensing[8] and 3D optical data storage[9] and so on. Specifically, higher-order nonlinear absorption (for example, 4PA or 5PA) are especially desirable for imaging[10]. For such applications, materials with large MPA cross-sections ($\sigma_n$) and high photoluminescence (PL) quantum yield (PLQY, $\eta$) are required, hence the parameter known as action cross-section (that is, $\eta \times \sigma_n$). Suitable media include: conventional inorganic semiconductor nanocrystals (NCs)[11–13], organic chromophores[14–16], polymers[17,18], metal complexes[19] and biomolecules[20]. Among them, the former exhibit exceptional performances with relatively large two- and three-photon action cross-sections (($\eta\sigma_2$)$_{max}$ $\sim 10^4$ GM and ($\eta\sigma_3$)$_{max}$ $\sim 10^{-76}$ cm$^6$s$^2$ photon$^{-2}$)[10–16,21,22]. Nonetheless, it is non-trivial to extend to 4PA/5PA in conventional semiconductor NCs (for example, II–VI) and organics because of their relatively smaller $\eta\sigma_{n\geq4}$, thereby requiring large intensities that can easily cause material damage. Hence, reports on four-photon-excited upconversion emission are few and far between[23,24]. To the best of our knowledge, five-photon-excited upconversion fluorescence from semiconductor NCs has yet to be demonstrated. Such endeavour for organic chromophores necessitates careful design and synthesis (for example, ($E$)-3-(4-(2-(1-hexyl-4-methyl-1$H$-imidazol-5-yl) vinyl) pyridinium-1-yl) propyl sulfate)[10].

Organic–inorganic halide perovskites, which demonstrated superior optoelectronic properties for photovoltaics and light emission[25,26], have recently demonstrated promising nonlinear optical properties[27–31]. These include: strong second-harmonic generation in organic/inorganic germanium perovskite compounds[27]; 2PA and two-photon-excited fluorescence in CH$_3$NH$_3$PbBr$_3$ (MAPbBr$_3$) perovskite bulk crystal at 800 nm (ref. 28); 2PA/3PA in CH$_3$NH$_3$PbI$_3$ (MAPbI$_3$) perovskite bulk crystal at three discrete wavelengths[29]; and large $\eta\sigma_2 \sim 10^6$ GM (at 800 nm) in cubic colloidal CsPbBr$_3$ NCs ($\sim 2$ orders larger than conventional semiconductor NCs[30,31]). Nonetheless, detailed understanding into these low-order nonlinear properties is still severely lacking (for example, the broad spectral dependence of 2PA/3PA cross-sections; the influence of the organic or inorganic A cation (that is, CH$_3$NH$_3^+$ or Cs$^+$) on the MPA). We also note an emerging controversy over the contrasting nonlinear optical behaviour for CsPbBr$_3$ NCs where both saturable absorption[32] and 2PA[30,31] at 800 nm were reported (Supplementary Table 1). Critically, their higher-order nonlinear optical properties (that is, 3-, 4-, 5PA) are presently unknown.

One viable approach to enhancing $\eta\sigma_n$ in quantum confined NCs is through the core–shell structure that permits/facilitates: effective passivation of non-radiative surface traps[33,34]; antenna-like effect[35]; photoinduced screening of the internal field[34,36]; and/or local field effect[34,37]. We recently realized highly luminescent (PLQY $\sim 92\%$) type I core–shell multidimensional perovskite MAPbBr$_3$/(OA)$_2$PbBr$_4$ NCs comprising of a wider bandgap 2D-layered perovskite ((OA)$_2$PbBr$_4$) shell encapsulating a 3D MAPbBr$_3$ NC core[38]. These novel halide perovskite NCs possesses the necessary criteria discussed above (as detailed in the Discussion section) that will open exciting new prospects for

tuning the higher-order nonlinear optical effects (that is, 4PA/5PA).

Herein, we reveal that the family of colloidal halide perovskite (that is, CsPbBr$_3$, MAPbBr$_3$ and core–shell MAPbBr$_3$/(OA)$_2$PbBr$_4$) NCs possesses ultralarge MPA properties, with five-photon action cross-sections ($\eta\sigma_5 \sim 10^{-136}$ cm$^{10}$s$^4$ photon$^{-4}$) that are at least 9 orders larger than the record values of designer organic molecules ($\eta\sigma_5 \sim 10^{-145}$ cm$^{10}$s$^4$ photon$^{-4}$) reported recently[10]. The MPA cross-sections (2-, 3-, 4- and 5PA) and their spectral dependences over the wide wavelength range of 675–2,300 nm are meticulously established. Importantly, multidimensional core–shell halide perovskite NCs with superior PL stability afford enhanced tunability of their MPA properties that would enable fresh approaches for multiphoton imaging applications.

## Results

**Giant 5PA from perovskite colloidal NCs.** These halide perovskite colloidal NC samples (in toluene) were synthesized using various solution-processed approaches. Briefly, CsPbBr$_3$ NCs (of side $\sim 9$ nm) were fabricated using a modified one-step technique[39], while both the core-only MAPbBr$_3$ NCs (diameter $\sim 8$-9 nm) and the core–shell MAPbBr$_3$/(OA)$_2$PbBr$_4$ NCs (diameter $\sim 9$–10 nm) were synthesized using the ligand-assisted reprecipitation method[38]—see Methods and Supplementary Information for more details of the synthesis and the transmission electron microscopy images. The core–shell MAPbBr$_3$/(OA)$_2$PbBr$_4$ NCs possess a type-I conduction and valence band edge alignment between the core and shell—see schematic in Figure 1a, as validated by both photoelectron spectroscopy in air measurements and theoretical calculations (Supplementary Figures 1,2 and Supplementary Methods). The linear optical properties (that is, 1PA and one-photon-excited PL—Supplementary Figure 1) of these halide perovskite NCs show clear signatures of excitonic and quantum confinement effects, where the larger blueshift in CsPbBr$_3$ NCs stems from its larger Bohr diameter ($d_B \sim 7$ nm) compared to MAPbBr$_3$ NCs ($d_B \sim 4$ nm), in agreement with literature reports (see Supplementary Methods for more details). Their PLQY are $\sim 84\%$, $\sim 92\%$ and $\sim 55\%$ for MAPbBr$_3$, MAPbBr$_3$/(OA)$_2$PbBr$_4$ and CsPbBr$_3$ NCs, respectively (see Supplementary Methods for details of the PLQY characterization). Figure 1b shows the five-photon-excited (that is, at 2,100 nm wavelength) excitonic PL emission (centred around 520 nm) from these samples with excitation fluence range $\sim 3.30$–5.61 mJ cm$^{-2}$ (inset) and their quintic excitation fluence dependence (inset). The 5PA excitation process via virtual energy levels is schematically illustrated in Figure 1c. Figure 1d shows the ultralarge $\eta\sigma_5$ (and their corresponding $\sigma_5$ values (inset)) obtained through employing the $\sigma_2$ values at 800 nm measured by Z-scan[40] (Supplementary Figures 3,4 and Supplementary Notes) as a standard and applying the calculation equations (see Supplementary Notes for details)[41] for these halide perovskite colloidal NC samples over the infrared excitation wavelengths of 2,050–2,300 nm. Record $\eta\sigma_5 \sim 10^{-136}$ cm$^{10}$s$^4$ photon$^{-4}$ values are achieved with MAPbBr$_3$/(OA)$_2$PbBr$_4$ NCs > CsPbBr$_3$ NCs > MAPbBr$_3$ NCs (see Table 1), highlighting the significance of the (OA)$_2$PbBr$_4$ shell for enhancing the nonlinear optical absorption action cross-sections. All three samples follow a similar spectral dependence with $\eta\sigma_5$ for MAPbBr$_3$/(OA)$_2$PbBr$_4$ NCs $\sim 6$–8 times larger than MAPbBr$_3$ NCs (see Table 1).

The 2PA cross-sections ($\sigma_2$ values at 800 nm) of the colloidal NCs (measured using the open-aperture Z-scan technique) were employed as a standard for multiphoton excited PL (MEPL)

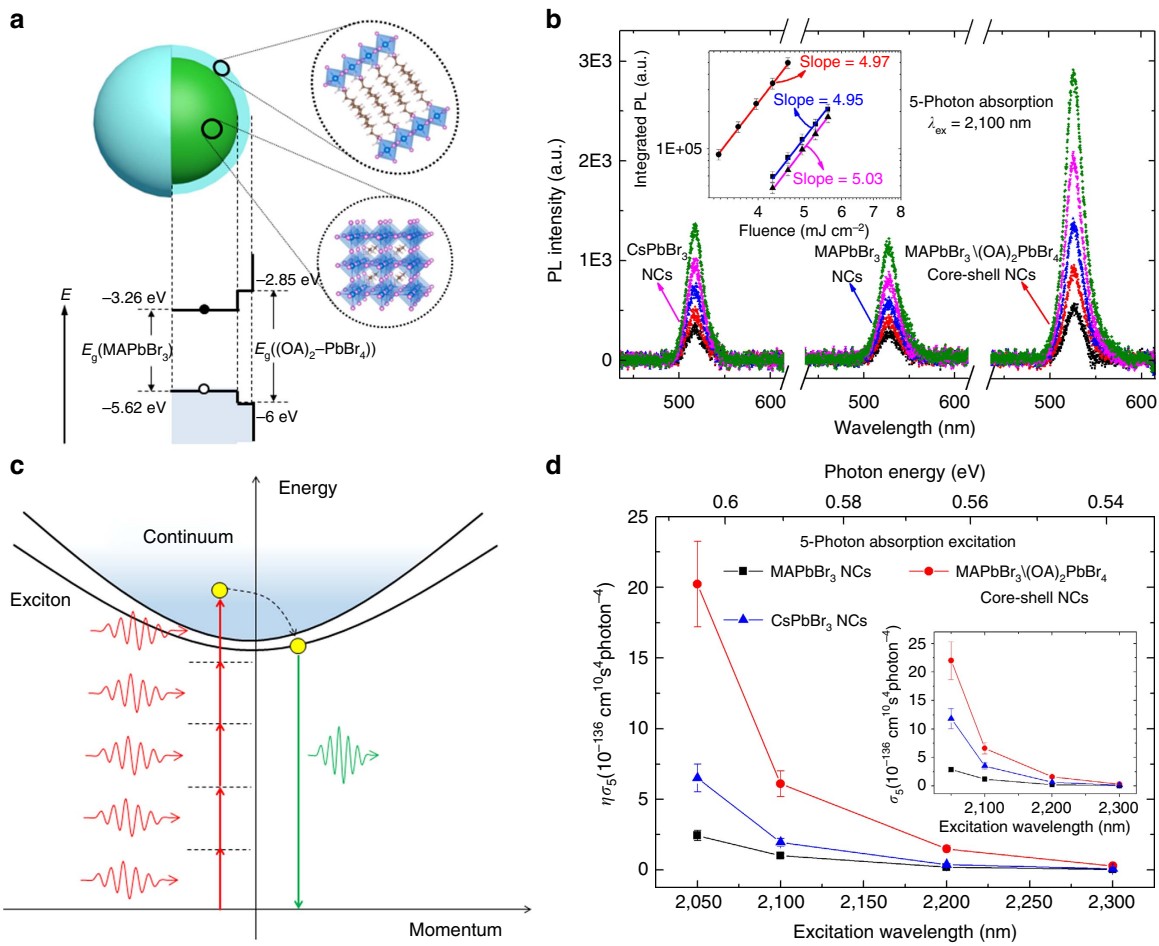

**Figure 1 | Type I core–shell multidimensional perovskite NCs and their giant five-photon action cross-sections ($\eta\sigma_5$). (a)** Schematic illustrating the core–shell multidimensional perovskite NCs with 3D MAPbBr$_3$ as core and 2D (OA)$_2$PbBr$_4$ as shell, and their type-I energy level alignment. **(b)** 5PPL spectra from core-only MAPbBr$_3$ NCs ($\sim$2.0 μM in toluene), core–shell MAPbBr$_3$/(OA)$_2$PbBr$_4$ NCs ($\sim$2.1 μM in toluene) and CsPbBr$_3$ NCs ($\sim$1.0 μM in toluene), with femtosecond laser excitation at 2,100 nm. Inset shows the quintic dependence on the excitation fluence of the spectrally integrated PL intensity. **(c)** Schematic illustrating the 5PPL process in perovskite NCs. **(d)** Five-photon action cross-section ($\eta\sigma_5$) spectra of the MAPbBr$_3$, MAPbBr$_3$/(OA)$_2$PbBr$_4$ and CsPbBr$_3$ NCs. Error bars indicate experimental uncertainty of $\pm$15%.

**Table 1 | Multiphoton action cross-sections $\eta\sigma_n$ of lead bromide perovskite NCs over their respective wavelength range*.**

| Multiphoton excitation | Two-photon $\eta\sigma_2$ ($10^6$ GM) (675–1,000 nm) | Three-photon $\eta\sigma_3$ ($10^{-74}$ cm$^6$s$^2$ photon$^{-2}$) (1,050–1,500 nm) | Four-photon $\eta\sigma_4$ ($10^{-104}$ cm$^8$s$^3$ photon$^{-3}$) (1,550–2,000 nm) | Five-photon $\eta\sigma_5$ ($10^{-136}$ cm$^{10}$s$^4$ photon$^{-4}$) (2,050–2,300 nm) |
|---|---|---|---|---|
| *Perovskite NCs* | | | | |
| MAPbBr$_3$ | 0.41 ± 0.06–5.2 ± 0.8 | 0.33 ± 0.05–2.7 ± 0.4 | 0.036 ± 0.005–3.0 ± 0.5 | 0.039 ± 0.006–2.4 ± 0.4 |
| MAPbBr$_3$/(OA)$_2$PbBr$_4$ | 3.0 ± 0.4–37 ± 6 | 2.5 ± 0.4–22 ± 3 | 0.21 ± 0.03–24 ± 4 | 0.29 ± 0.04–20 ± 3 |
| CsPbBr$_3$ | 1.0 ± 0.2–13 ± 2 | 0.38 ± 0.06–8.0 ± 1.0 | 0.07 ± 0.01–7.0 ± 1.0 | 0.09 ± 0.01–6.5 ± 1.0 |

NC, nanocrystal; MEPL, multiphoton-excited photoluminescence.
*The experimental error ±15% stems mainly from the uncertainty in fluctuation of input laser pulse energy and determination of laser beam characteristics such as pulse duration and minimum beam waist, which are essential for both open-aperture Z-scan and MEPL measurements.

measurements at different wavelengths (see 'Methods' section and Supplementary Notes for more details). Our measured $\sigma_2$ values (from Z-scan) for CsPbBr$_3$ NCs at 800 nm agrees well with literature reports[30,31]. The consistency of the ratio of the $\sigma_2$ values (for MAPbBr$_3$, CsPbBr$_3$ and MAPbBr$_3$/(OA)$_2$PbBr$_4$ NCs at 800 nm) obtained from Z-scan (Supplementary Figure 4) with the two-photon-excited PL measurements (Supplementary Figures 5, 6a and 7a) provides further validation of our approach. Moreover, the good agreement of the measured 3PA cross-sections with those acquired from open-aperture Z-scan measurements at 1,050 and 1,100 nm (see Supplementary

Figures 6b,7b,8 and 9 and Supplementary Notes for more details) further confirms that the MPA cross-sections have been properly measured with the MEPL technique. Similar to Figure 1b, MEPL spectra under 2-, 3- and 4-photon excitation for the NCs at 800, 1,200 and 1,600 nm with excitation fluence ranging from $\sim$0.33 to 1.16, $\sim$0.66 to 1.82 and $\sim$1.16 to 2.31 mJ cm$^{-2}$ (that is, higher upconversion PL intensities in the lower-order multiphoton processes even at lower excitation fluence), respectively, are shown in Supplementary Figures 6a–c, and illustrated schematically in Supplementary Figure 6d. Supplementary Figures 6a–c insets clearly show the nearly

quadratic, cubic and quartic dependences of the spectrally integrated PL intensity on excitation fluence for the 2-, 3- and 4-photon processes with excitation fluence ranging from $\sim 0.33$ to $1.16, \sim 0.66$ to $1.82, \sim 1.16$ to $2.31 \, mJ \, cm^{-2}$, respectively. The comparison between the normalized one-photon-excited PL spectra and MEPL spectra of the NCs are displayed in Supplementary Figure 10. As illustrated in Supplementary Figure 10, the slight redshift of the MEPL spectra with respect to one-photon counterpart has been well reported in traditional semiconductor NCs[42] and can be ascribed to the reabsorption effect and size inhomogeneity[43,44]. Further excitation fluence-dependent MEPL measurements at wavelengths ranging from 675 to 2,300 nm were also performed. Photographs in Supplementary Figure 11 clearly demonstrate the frequency-upconverted PL from NCs when they were irradiated with infrared femtosecond laser pulses. Furthermore, the direct comparison between the five-photon excited upconversion PL from $MAPbBr_3/(OA)_2PbBr_4$ NCs and that from R6G having the same concentration and under the same experimental conditions exemplify the superior 5PA properties of the $MAPbBr_3/(OA)_2PbBr_4$ NCs (see Supplementary Figure 12 and Supplementary Notes for more details).

**Multiphoton action cross-section spectra in perovskite NCs.** Figure 2a shows a summary of the excitation wavelength dependence of the slopes (that is, orders of MPA processes) for these halide perovskite NCs spanning from 675 to 2,300 nm. The slopes of PL from the NCs are around 2 in the wavelength range

of 675–1,000 nm, clearly indicating 2PA. As the excitation wavelength increases to 1,050–1,500 nm, the slopes increase to around 3—revealing a switch of the excitation mechanism to 3PA. 4PA processes (with slopes around 4) dominate as the excitation wavelength is further increased to the range of 1,550–2,000 nm. In the long wavelength range 2,050–2,300 nm, the slopes are around 5, indicating the dominance of 5PA processes. Due to the NCs' size inhomogeneity, $\eta \sigma_n$ at the wavelength boundaries contains an admixture of contributions from both the lower- and higher-order MPA process (that is, 2/3PA, 3/4PA and 4/5PA and so on). Instead of a sharp transition at the wavelength boundaries (of 1,050, 1,550 and 2,050 nm), the slope deviates from the integer value by $\sim 20\%$. Figure 2b–d shows a direct comparison of the NCs' MEPL brightness (that is, $\eta \sigma_n$) for lower orders, $n = 2, 3$ and 4, over the wavelength range 675–2,000 nm. Similar to $\eta \sigma_5$, their spectral dependences exhibit a general overall decreasing trend with increasing wavelength. Their corresponding $\sigma_n$ values obtained through applying the calculation equations (see Supplementary Notes for details) and employing the $\sigma_2$ values at 800 nm measured by Z-scan as a standard are given in Figure 2b–d insets. In all these cases, the core–shell $MAPbBr_3/(OA)_2PbBr_4$ NCs possess $\sim 6$–8 times larger $\eta \sigma_n$ values than the core-only $MAPbBr_3$ NCs. The $\eta \sigma_n$ spectra suggest that the growth of the 2D $(OA)_2PbBr_4$ shell over $MAPbBr_3$ NCs only enhances their MPA and PLQY, while hardly changing their spectral response.

Table 1 shows the elucidated $\eta \sigma_n$ for $MAPbBr_3$, $MAPbBr_3/(OA)_2PbBr_4$ and $CsPbBr_3$ NCs. Comparatively, the $\eta \sigma_2$ values of

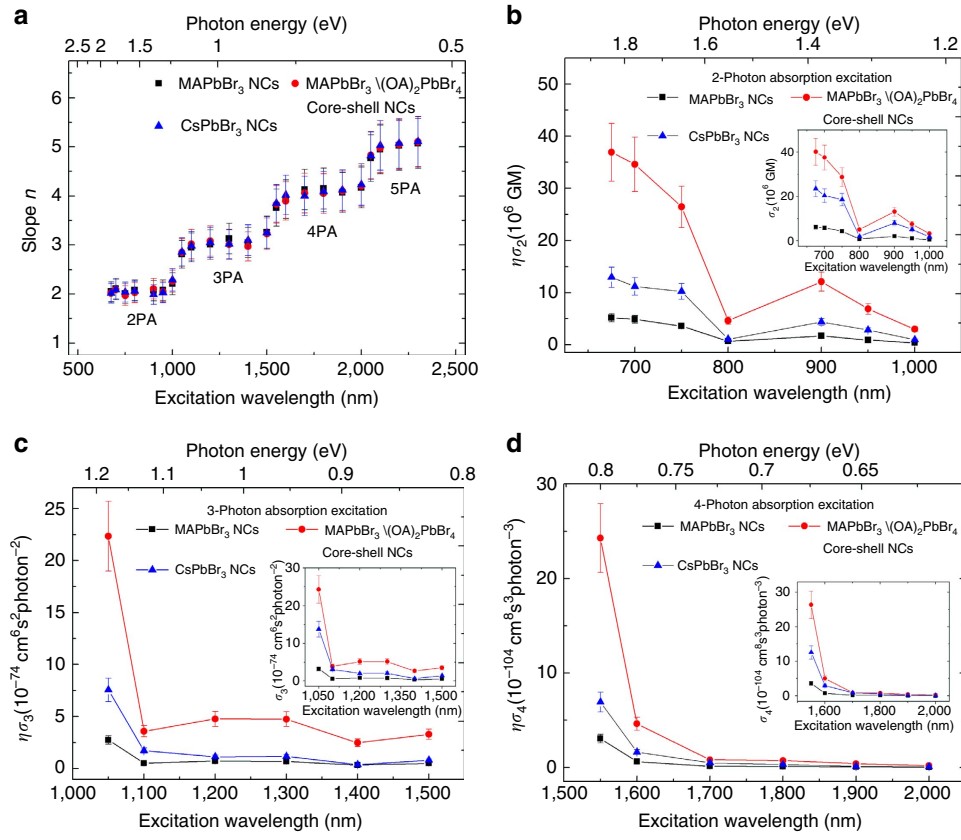

**Figure 2 | Excitation fluence dependence of multiphoton-excited upconversion PL(slopes) and action cross-sections as a function of excitation wavelengths.** (**a**) Slopes $n$ plotted as a function of laser excitation wavelength (photon energy), where $n$ is defined as the excitation fluence dependence of the MPPL signal that is proportional to (excitation fluence)$^n$. (**b**) Two-photon action cross-section ($\eta \sigma_2$) spectra of the $MAPbBr_3$, $MAPbBr_3/(OA)_2PbBr_4$ and $CsPbBr_3$ NCs in the wavelength range 675–1,000 nm. (**c**) Three-photon action cross-section ($\eta \sigma_3$) spectra of the perovskite NCs in the range 1,050–1,500 nm. (**d**) Four-photon action cross-section ($\eta \sigma_4$) spectra of the perovskite NCs in the range 1,550–2,000 nm. Insets in (**b**–**d**) show the corresponding spectral dependence of MPA cross-sections ($\sigma_n$) of the perovskite NCs. Error bars indicate the experimental uncertainty of $\pm 15\%$.

these lead bromide perovskite NCs are 1–2 orders larger than that of large size CdSe/CdS dot in rod heterostructures (39 or 180 nm length CdS nanorod—Supplementary Table 2)[35,45], and are 2–3 orders larger than traditional inorganic semiconductor NCs and organic chromophores (Supplementary Table 2). For $\eta\sigma_3$ values, the NCs are 1–2 orders larger than the strongly (or overly) excited ZnSe/ZnS core–shell NCs[46] (where three-photon-excited intraband absorption has a significant contribution[47]) and the large CdSe/CdS dot-in-rod heterostructure (39 nm length CdS nanorod)[48] (Supplementary Table 3).

Our lead bromide perovskite NCs exhibit $\eta\sigma_3$ values 2–3 orders of magnitude higher than the best performing conventional inorganic semiconductor NCs (that are not overly excited and with size $\leq 10$ nm) and organic molecules (see Supplementary Table 3). For the higher-order $\eta\sigma_4$ values, our NCs are $\sim 3$–5 orders larger than the best reported results of organic chromophores[16,23,24] (see Supplementary Table 4). However, there are only limited research efforts devoted to studying the four-photon process in conventional inorganic semiconductor NCs. Amazingly, the $\eta\sigma_5$ values of our NCs are $>9$ orders larger than specially designed organic molecules[10] ($\eta\sigma_5 \sim 10^{-145}$ cm$^{10}$ s$^4$ photon$^{-4}$) (see Supplementary Table 5), which is a record for semiconductor NCs. Although the 'Luttinger-Kohn' and 'Pidgeon-Brown' models within the $\mathbf{k} \cdot \mathbf{p}$ approach have been successfully applied to conventional metal-chalcogenide NCs[49–51] to model their 2PA/3PA spectral dependences and estimate their $\sigma_2/\sigma_3$, such approach cannot be directly applied to lead bromide perovskite NCs. This is because these methods are only suitable for the two-photon transitions from s-type valence band to p-type conduction band as in the former, unlike the all p-type valence and conduction bands in the latter. The theoretical study on the MPA properties of the lead bromide perovskite NCs will be the focus of a future work.

## Discussion

The family of lead bromide perovskite NCs possesses much higher $\eta\sigma_n$ values than those of traditional inorganic semiconductor NCs and organic chromophores (Table 1 and Supplementary Tables 2–5), highlighting their potential for nonlinear optics and bioimaging applications, such as optical limiting, 3D microscopy for deep tissue imaging and sensing. Apart from having high PLQY (that is, $\eta \sim 84\%$, 92% and 55% for MAPbBr$_3$, MAPbBr$_3$/(OA)$_2$PbBr$_4$ and CsPbBr$_3$ NCs, respectively), these lead bromide perovskite NCs also possess giant $\sigma_n$ values, which stems mainly from the intrinsic strong MPA of lead bromide perovskites. Among our perovskite NCs, a combination of effects like the relatively stronger quantum confinement in CsPbBr$_3$ NCs ($d_B = 7$ nm) and/or the influence from Cs$^+$ cation give rise to their larger MPA cross-sections than MAPbBr$_3$ NCs (weaker confinement). By growing a (OA)$_2$PbBr$_4$ shell over the MAPbBr$_3$ core, the PLQY can be enhanced to $\sim 92\%$ and the photostability can be improved, as demonstrated here and in ref. 38. Most importantly, the presence of the (OA)$_2$PbBr$_4$ shell enhances the $\sigma_n$ by almost an order compared to the core-only MAPbBr$_3$ NCs across all wavelengths from 675 to 2,300 nm. Given that both our MAPbBr$_3$ and CsPbBr$_3$ NCs are in the weak confinement regime ($d_B \sim 4$ nm $\ll 8$–9 nm diameter for MAPbBr$_3$ and $d_B \sim 7$ nm $< 9$ nm side for the cubic CsPbBr$_3$), it is possible to enhance these $\sigma_n$ values further through even smaller strongly confined MAPbBr$_3$ NCs, their core–shell counterparts (that is, NCs' diameter $< 4$ nm) and CsPbBr$_3$ NCs (that is, NCs' side $< 7$ nm). However, it is presently extremely challenging to synthesize small MAPbBr$_3$ (refs 52,53) and CsPbBr$_3$ NCs[54,55] with high crystalline quality, low surface defects, uniform size

distribution and relatively high reaction yield, which are needed to conduct detailed investigations of the quantum confinement effect on the MPA of lead bromide perovskite NCs. The all-inorganic CsPbBr$_3$ NCs were prepared via the hot-injection method[39] at temperatures between 140–180 °C. The NC size is determined by the reaction temperature, where larger NCs are obtained at higher temperature. However, below 140 °C, nanoplatelets (not NCs) are formed instead[54,55]. Moreover, at the current stage, despite MAPbBr$_3$ NCs with small sizes can be obtained[52,53], various problems associated with high density surface defects, low sample crystalline quality, low reaction yield and non-uniform size distribution need to be further addressed[52,53]. Presently, there have been no reports on the synthesis of small-sized core–shell MAPbBr$_3$/(OA)$_2$PbBr$_4$ NCs. Therefore, significant advances in perovskite NCs synthesis are needed before this question on the detailed dependence of MPA on the quantum confinement effect could be answered.

Previous studies on conventional inorganic semiconductor NCs have shown that the 2PA and 3PA cross-sections can be enhanced by an outer shell covering through: surface passivation effects[33,34]; antenna-like effect[35]; photoinduced screening of the internal field[34,36]; and/or local field effects[34,37]. To establish the origins for the $\sigma_n$ enhancement of the core–shell MAPbBr$_3$/(OA)$_2$PbBr$_4$ NCs and to gain more insights into the dynamics of excited photocarriers, time-resolved PL measurements using single (400 nm) and multiphoton (for example, 2P (800 nm), 3P (1,200 nm), 4P (1,600 nm) and 5P (2,100 nm)) excitation were performed. Figure 3a–c shows the one-photon- and multiphoton-excited time-resolved PL decay lifetimes for the MAPbBr$_3$, MAPbBr$_3$/(OA)$_2$PbBr$_4$ and CsPbBr$_3$ NCs, respectively. For the respective MAPbBr$_3$ and CsPbBr$_3$ NCs, their PL decay curves are almost invariant for one- and multiphoton excitation (Figure 3a,c), indicating excitation via virtual states to the same lowest excited state in these perovskite NC systems.

The increased PL lifetimes (from one- and multiphoton excitation—Figure 3b) together with the enhanced PLQY and stability[38] in MAPbBr$_3$/(OA)$_2$PbBr$_4$ NCs compared to the core-only MAPbBr$_3$ NCs points to the effective surface passivation provided by the (OA)$_2$PbBr$_4$ shell that reduces the nonradiative surface traps, thus increasing the multiphoton transition probability[33,34]. Furthermore, the longer one- and multiphoton excited PL lifetime, the good spectral overlap between the shell emission and the core excitation (as revealed in Supplementary Figure 13), as well as the intimate proximity between the core and shell strongly suggest the presence of non-radiative Förster-type energy transfer from the shell to the core through an antenna-like effect. This is another plausible origin for the enhanced $\sigma_n$ in the core–shell MAPbBr$_3$/(OA)$_2$PbBr$_4$ NCs[19,35]. The 2PA property of the (OA)$_2$PbBr$_4$ shell revealed by open-aperture Z-scan measurements in Supplementary Figure 14 further supports this interpretation. The non-radiative energy transfer from the shell to core enhances the PL from the MAPbBr$_3$ NCs core, resulting in an overall increase of the PL lifetime[19,35]. The longer multiphoton excited PL lifetime compared to the one-photon excited PL lifetime in MAPbBr$_3$/(OA)$_2$PbBr$_4$ NCs (Figure 3b) indicates a larger contribution from this non-radiative energy transfer channel to the resulting core PL, suggesting a more effective non-radiative energy transfer under nonlinear optical excitation[19,35].

Next, the relatively large lattice mismatch between the core MAPbBr$_3$ (ref. 56) and shell (OA)$_2$PbBr$_4$ (ref. 57) could also induce intrinsic piezoelectric polarization charges at the core–shell interface that result in an internal electric field. This indicates that the photoinduced screening of the internal field by the photoexcited electron–hole pairs could be another possible

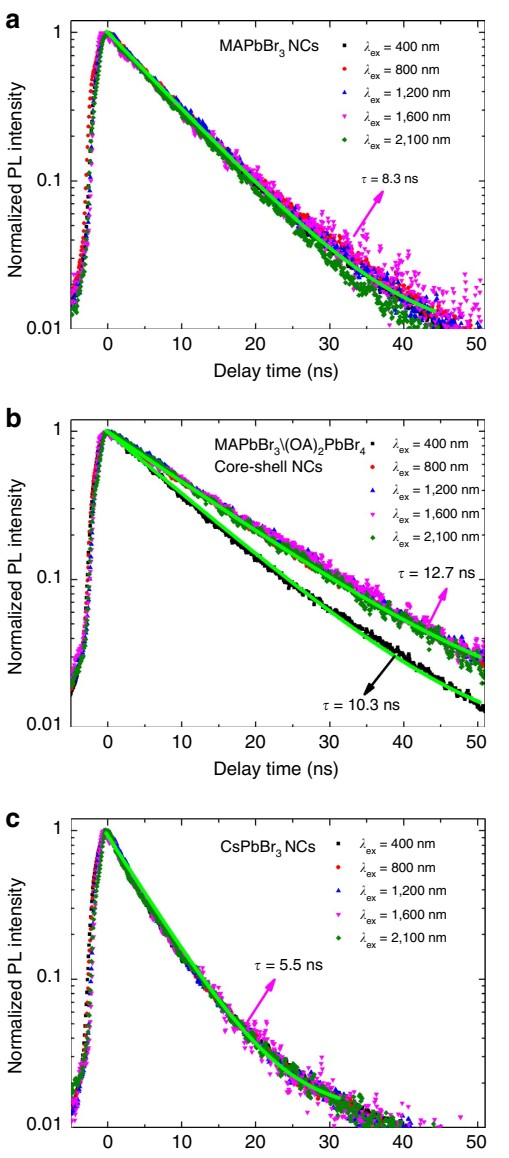

**Figure 3 | Comparison between the PL decay traces of the perovskite NCs under various multi-photon excitations.** (**a**) One- and multiphoton-excited PL decay curves in MAPbBr$_3$ NCs. (**b**) Longer one- and multiphoton-excited PL decay lifetimes are obtained for MAPbBr$_3$/(OA)$_2$PbBr$_4$ NCs than for MAPbBr$_3$ NCs. (**c**) Shorter PL decay lifetimes in CsPbBr$_3$ NCs under one- and multiphoton excitation.

origin for the enhanced $\sigma_n$ in MAPbBr$_3$/(OA)$_2$PbBr$_4$ NCs[34,36]. Furthermore, the dielectric confinement (local field effect) resulting from the relatively large difference between the dielectric constants of the core MAPbBr$_3$ (ref. 58) and shell (OA)$_2$PbBr$_4$ (ref. 59) could be another factor[34,37]. Therefore, it is likely that an interplay of various factors arising from the effective surface passivation, the photoinduced screening of the internal field, the nonradiative energy transfer from the shell to the core through antenna-like effect, the local field effect or its combination gives rise to the giant MPA cross-sections of these core–shell multidimensional halide perovskites NCs.

The outstanding higher-order nonlinear optical properties of the lead bromide perovskite colloidal NCs (particularly the highly efficient five-photon excited upconversion fluorescence in the multidimensional core–shell perovskite NCs (for example,

MAPbBr$_3$/(OA)$_2$PbBr$_4$)) indicate their great potential for developing next-generation highly efficient, sensitive multiphoton imaging applications with unmatched imaging depth and resolution. Although the intrinsic low stability of halide perovskite NCs in polar solvents (such as water) and their cytoxicity from the Pb$^{2+}$ ion could be an issue for potential applications in multiphoton bioimaging, such technical challenges could be circumvented through the following three possible encapsulation approaches:

The first is encapsulation with a SiO$_x$/SiO$_2$ inert shell/matrix. Utilizing SiO$_x$/SiO$_2$ either as an additional shell or as a medium to embed NCs will help to enhance the stability as well as reduce the cytoxicity. Such approach has been successfully applied to traditional inorganic semiconductor NCs[60–64] for efficient bioimaging applications. Very recently, encapsulating lead bromide NCs into a SiO$_x$ spherical matrix (~150 and 470 nm in diameter) have been realized by Huang et al.[65] This provides a crucial proof-of-concept on the viability of the approach, although more work is required to achieve finer control of SiO$_x$/SiO$_2$ encapsulation for each perovskite NC.

The second is encapsulation with ligands. Through utilizing 3-aminopropyltriethoxysilane (APTES) as a branched capping ligand to replace the conventional straight-chained ligands, Luo et al.[66] achieved enhanced stability in the APTES-coated MAPbBr$_3$ NCs. Such APTES-coated MAPbBr$_3$ NCs show an increased stability in 2-propanol. Furthermore, the available silane group allows for additional sol–gel reactions, which may further increase the stability of the NCs.

The third is encapsulation with solid lipid structures. More recently, Gomez et al.[67] reported the application of solid lipid structures to encapsulate CsPbBr$_3$ NCs that were stable up to 2 months in water, albeit at a low PLQY (around 11%) likely due to initial water degradation. Further work is needed to circumvent the initial degradation and realize water-stable NCs at high PLQY.

These examples clearly show that the potential stability and cytotoxicity issues of halide perovskite NCs in aqueous media for multiphoton imaging applications could, in fact, be overcome through a judicious choice of the encapsulation approach and careful optimization. Through tuning the emission wavelengths to the infrared (for example, using CsSnI$_3$ NCs), these halide perovskite colloidal NCs could enable even deeper imaging for deep-tissue bioimaging, where both penetration of the incident photons and extraction of the emitting photons are essential.

In retrospect, our findings reveal that the family of halide perovskite colloidal NCs possesses outstanding higher-order nonlinear optical properties (for example, highly efficient five-photon-excited upconversion fluorescence), which is unprecedented for semiconductor NCs. Particularly, overcoating the 3D perovskite nanocrystals (for example, MAPbBr$_3$) with a 2D perovskite shell (for example, core/(OA)$_2$PbBr$_4$ shell) to form multidimensional core–shell perovskite nanocrystals offers amazing five photon action cross-sections that are at least 9 orders larger ($\eta\sigma_5 \sim 10^{-136}$ cm$^{10}$s$^4$ photon$^{-4}$) than state-of-the-art specially designed organic molecules. Our work aptly demonstrates that this new family of perovskite nanocrystals is a promising class of nonlinear optical materials for developing next-generation multiphoton imaging applications with unmatched imaging depth, sensitivity and resolution.

## Methods

**Sample preparation.** The methylammonium lead bromide perovskite (MAPbBr$_3$) nanocrystals were synthesized using the recently developed ligand-assisted reprecipitation strategy[38]. A mixture of 0.16 mmol methylammonium bromide (MABr), 0.2 mmol lead bromide (PbBr$_2$) was first dissolved in 5 ml dimethylformamide (DMF) in a glass vial. Then, 50 µl oleylamine (OAm) and 0.5 ml oleic acid

(OAc) were added into the above DMF solution to make the precursor solution. 250 microlitres of the as-prepared precursor solution was then swiftly injected into 5 ml of toluene kept at 60 °C in a round bottom glass flask, and stirred vigorously for 5 min. The MAPbBr$_3$ nanocrystals were formed as confirmed by the change of the solution colour into green. The mixture was transferred into a centrifuge tube to be centrifuged at a relative centrifugal force of about 6738g (Model: Eppendorf 5804R, rotor FA-45-6-30, rotor radius 12.3 cm). For 10 min to purify the nanocrystals. The supernatant containing the nanocrystals was collected for further investigations and the precipitate was discarded. More details can be found in ref. 38.

To fabricate the core–shell multidimensional perovskite nanocrystals MAPbBr$_3$/(OA)$_2$PbBr$_4$, octylammonium bromide was first synthesized utilizing octylamine and Hydrobromic acid (HBr). Then, methylammonium bromide and octylammonium bromide with molar ratio of 8:2 were added to the precursor solution. The core–shell multidimensional perovskite NCs MAPbBr$_3$/(OA)$_2$PbBr$_4$ were synthesized following the same synthetic procedures elaborated above. The core–shell NCs were formed through the co-precipitation of the mixed methylammonium and octylammonium bromide precursors. More details can be found in ref. 38.

In addition, the colloidal CsPbBr$_3$ nanocrystals with cubic shape and edge length ∼9 nm were synthesized following the reported modified one-step technique[39]. The steps are as follows: preparation of Cs-oleate precursor: 814 mg (2.5 mmol) of caesium carbonate (Cs$_2$CO$_3$) was mixed with 2.5 ml OAc and 40 ml 1-octadecene (ODE) in a 100 ml three-neck round bottom glass flask. At first, the mixed solution was dried by a vacuum pump for 1 h at 120 °C. Then, the reaction mixture was heated under nitrogen environment for another 1 h at 150 °C until all Cs$_2$CO$_3$ reacted with OA and becomes transparent. Cs-oleate precursor was preheated to 100 °C before injection. Synthesis of cubic caesium lead bromide NCs: 69 mg (0.188 mmol) of PbBr$_2$ was mixed in 5 ml ODE, 0.5 ml OAc and 0.5 ml oleylamine in a three-necked flask. The flask was dried under vacuum for 1 h at 120 °C and degassed with nitrogen gas for another 1 h at same temperature. Cs-oleate precursor of 0.4 ml was injected into the main reaction solution at 170 °C via syringe, yielding a bright green colloid NCs. The reaction was quenched with a ice bath for 30 s after the injection of the Cs-oleate precursor. The colloidal CsPbBr$_3$ NCs was transferred to centrifuge tube with some acetonitrile. The mixture was centrifuged at relative centrifugal force of about 7587 × g for 10 min and the precipitation was redispersed in toluene for further experiments.

**Sample characterization.** Structural characterization of the as-synthesized nanocrystals was performed using both transmission electron microscopy (Jeol JEM-2010) and the X-ray diffraction (XRD Bruker D8 Advance). The concentrations of perovskite nanocrystals in toluene solution were determined by the mass of lead, which were measured by the inductively coupled plasma optical emission spectrometry (ICP-OES Optima 8000; Perkin-Elmer).

1PA spectra of the perovskite nanocrystals in solution phase (toluene) were measured using a Shimadzu UV1800 Ultraviolet –Visible spectrophotometer. A Shimadzu RF-5301pc spectrofluorophotometer was employed to record the one-photon-excited PL spectra of the nanocrystals in toluene. Measurements of the absolute PLQYs of the perovskite nanocrystals were performed using an Ocean-optics USB4000 spectrometer with a BaSO$_4$-coated integrating sphere excited by laser beam at 400 nm.

**Multiphoton-excited PL and time-resolved PL measurements.** For multiphoton-excited frequency-upconverted PL measurements[41], a femtosecond amplified-pulsed laser system was used as the excitation source. The excitation laser pulses (∼50 fs, 1 kHz, 250–2,600 nm) were generated by an optical parametric amplifier (OperA-Solo, Coherent) pumped by a regenerative amplified femtosecond Ti:Sapphire laser system (∼50 fs, 800 nm, 1 kHz; Libra, Coherent). The Coherent Libra regenerative amplifier was seeded by a femtosecond Ti:Sapphire oscillator (∼50 fs, 80 MHz, Vitesse, Coherent). The temporal, spectral and spatial profiles of the applied excitation laser source at wavelengths 675–2,300 nm follow Gaussian distribution based on the specifications in the manufacturer's data sheets, and the pulse widths at different wavelengths are in the range of 50–60 fs. Moreover, the Gaussian distribution of the temporal, spectral profiles of the applied excitation laser source at wavelengths 675–2,300 nm was experimentally verified by the characterizations with a single-shot autocorrelator (High-Resolution Single-Shot Autocorrelator, Coherent) (see Supplementary Table 6 and Supplementary Methods), a frequency-resolved optical gating (Swamp Optics, UPM-8-50) (see Supplementary Figure 15 and Supplementary Methods), visible monochromator (Acton, Spectra Pro 2750i) coupled with CCD (Princeton Instruments, Pixis 100B) and infrared monochromator (Acton, Spectra Pro 2300i) coupled with liquid-nitrogen-cooled InGaAs infrared detector (Princeton Instruments, 7490–0001) (see Supplementary Figure 16 and Supplementary Methods). Additionally, the pulse widths at different wavelengths were measured to be in the range of (50.6 ± 5.1)–(62.9 ± 6.3) fs, highly consistent with the specifications in the data sheet (see Supplementary Figure 16 and Supplementary Methods). Moreover, through applying knife-edge scans along both the x- and y-directions of the cross-sectional planes of the laser beams, the 2D spatial profiles of the laser beams at wavelengths 675–2,300 nm were validated to follow a Gaussian distribution (see Supplementary Figure 17 and Supplementary Methods).

The optical parametric amplifier output was filtered by a suitable filter (long-pass filter cut at 750 nm for wavelength >800 nm, notch filter at 800 nm for wavelengths between 675 and 750 nm) to stop light at undesired wavelengths. A circular lens with focal length of 20 cm was applied to focus the laser pulses onto the perovskite nanocrystals in toluene solution contained in a 2-mm-thick quartz cuvette. The NC samples were placed 23.5 cm away from the lens (3.5 cm away from the focal point) to avoid the high excitation peak intensity on the samples and to have larger excitation area (thus larger frequency-upconverted PL signal). Two continuously variable neutral density filters were employed to control the incident energy of the laser pulses. The frequency-upconverted PL from the nanocrystals was collected at a backscattering angle of 150° utilizing a pair of lenses into an optical fibre which was coupled into a spectrometer (Acton, Spectra Pro 2750i) and detected by a charge-coupled device (Princeton Instruments, Pixis 100B). A short-pass filter cut at 650 nm was placed in front of the fibre to remove the scattered light at the excitation laser frequencies. For the time-resolved PL measurements, the one-photon- and multiphoton-excited PL emission were acquired by a Optronis Optoscope streak camera system with an ultimate temporal resolution of ∼10 ps.

**Open-aperture Z-scan measurements.** For the Z-scan measurements[40] at 800 nm, 1050 nm and 1100 nm excitations, a beam splitter was employed to divide the incident laser beam into two parts. The first part served as the reference and was directed into a reference power detector ($D_R$, RkP 465, Laser Probe). The other part functioned as the signal beam and was focus by a circular lens with 20 cm focus length onto a 1-mm-thick quartz cuvette filled with the toluene solution of perovskite nanocrystals. The transmitted signal laser beam through the nanocrystals was detected by a signal power detector ($D_S$, RkP 465, Laser Probe). Both the reference and signal detectors were coupled with a powermeter (RM6600, National Instrument), whose readings were exported by a Labview USB data acquisition (DAQ, National Instrument) to a computer. The sample was controlled by a linear motorized stage to travel back and forth along the propagation direction of the laser beam (z axis). The transmission of the signal beam through the sample was monitored while translating the sample through the focus, and the transmission was recorded as a function of the sample position (z). With the incident laser pulse energies kept at a constant level, the sample experiences various laser irradiance I(z) at different z-positions, giving rise to corresponding changes in transmission if the sample absorbs light nonlinearly. All optical measurements were performed at room temperature.

**Data availability.** The data that support the findings of this study are available from the corresponding author on reasonable request.

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

## Acknowledgements

Financial support from Nanyang Technological University start-up Grants M4080514 and M4081293; the Ministry of Education Academic Research Fund Tier 1 Grants RG184/14 and RG101/15, and Tier 2 Grants MOE2013-T2-1-081, MOE2014-T2-1-044 and MOE2015-T2-2-015; the NTU-A*STAR Silicon Technologies Center of Excellence Program Grant 11235100003 and from the Singapore National Research Foundation through the Singapore–Berkeley Research Initiative for Sustainable Energy (SinBeRISE) CREATE Program and the Competitive Research Program NRF-CRP14-2014-03 is gratefully acknowledged.

## Author contributions

T.C.S., W.C. and N.M. conceived the idea for the manuscript and designed the experiments. W.C. conducted the spectroscopic characterization. S.B. and S.A.V. prepared the samples and performed sample characterization. Q.X. performed the DFT studies. W.C., M.G., S.M., N.M. and T.C.S. analysed the data and wrote the manuscript. All authors discussed the results and commented on the manuscript at all stages. T.C.S. and N.M. led the project.

## Additional information

**Competing interests:** The authors declare no competing financial interests.

