## [Peer Review File · Nature Communications]

Reviewers' comments:

Reviewer #1 (Remarks to the Author):

Prof. Sum and colleagues report high-order nonlinear optical response in the emergent material of perovskite semiconductor nanocrystals by showing efficient PL emission induced by multi-photon absorption in the core-shell halide nanocrystals. As the first experimental evidence of efficient 5-photon process in semiconductor nanocrystals, the work is timely and interesting for the community studying NLO in semiconductor nanomaterials. The conclusions are well supported. Thus, I recommend publishing the paper after addressing some technical concerns summarized below.

1)The authors have emphasized multi-photon imaging as an important potential application for their findings. However, halide perovskite nanocrystals are sensitive to moisture. It seems the material is intrinsically instable in water where multiphoton imaging is conducted. Can the authors discuss on the potential approach to tackle this technical challenge?

2)The slope of the power dependence of PL on the incident power has been taken as a metric to distinguish the order of multi-photon processes with excitation of different wavelengths (Figure 2a). I guess the ranges of the excitation power are different for different wavelengths. It is likely that the higher-order processes dominate if the power is sufficiently high even with higher photon energy. The authors should include the information of power ranges for each wavelengths used in these measurements.

3)In the discussion part, the authors have included in-depth discussion on the enhancement of multi-photon process in core-shell nanocrystals with respect to that to core-only nanocrystals, which is good. However, the intrinsic mechanism why perovskite semiconductor nanocrystals have a dramatically enhanced MPA has not been explained. Moreover, the authors note the possible involvement of quantum confinement in nanocrystals. Is it possible to show a more detailed dependence of 5PA to explain whether the excitonic effect relevant to the quantum confinement effect is critical?

4)In Figure 1(c) & Figure S6, the authors have schematically illustrated the transition induced by MPA in momentum space. In general, the momentum of a photon is negligible in compared to that of an electron. The processes of efficient MPA and PL emission here are likely to be direct transitions without involvement of any phonons. I would suggest the authors to draw these transitions vertically to avoid potential misunderstandings.

Reviewer #2 (Remarks to the Author):

Chen et al. reported the high-order nonlinear multiphoton upconverting luminescence (UCL) in the family of halide perovskite colloidal semiconductor nanocrystals (NCs). Specifically, large 5-photon action cross-sections ($\eta\sigma^5$) that is at least 9-orders larger than state-of-the-art specially-designed organic molecules was first demonstrated. It was revealed that the multi-dimensional core-shell nanostructure design contributed significantly to such large value of $\eta\sigma^5$. The results are convincing and very important in the fields of both perovskite NCs and multiphoton UCL. The manuscript is well organized and scholarly written in a very high quality. Therefore, I recommend the publication of this manuscript in Nat. Commun. after the following minor revisions:

1. The authors stated that higher-order nonlinear UCL are more favorable for bioimaging (e.g., high spatial confinement, long penetration depth, and low biological damage). However, the low efficiency of the high-order nonlinear UCL may limit its practical bioapplications. Meanwhile, the Pb²⁺ ion in the matrix of the perovskite NCs is toxic, and the halide perovskite NCs have a poor stability and are incompatible with biosystems in view of their hydrophobic nature. The authors are suggested to address this concern with more discussion or perspectives.

2. For the potential bioapplication of UCL, some most relevant papers might be included in the background Introduction.

Reviewer #3 (Remarks to the Author):

According to authors, the manuscript reports for the first time five-photon absorption on NC's "5-P-excited upconversion ... has yet to be demonstrated.". Also, the authors reported that the value is about 9 orders higher than the best organic molecule reported on the literature. The main subject of the manuscript is about the higher five photon absorption measured and determined by the authors. As the authors said, five photons, or multi-photon absorption were already studied in a large number of organic and inorganic materials, as can be seen on the references cited on the manuscript. But the fact of have an enhancement of 9 order compared to the best organic molecules motivates NC, showed on this manuscript, to be used on multiphoton fluorescence microscopy.

I could say that the results could be interesting to the field. However, the main point that I will question is the method used to confirm the high order absorption cross-section measured.

Of course, the authors carefully demonstrated that the fluorescence processes for different wavelengths regions are dominated by different order of multiphoton absorption, easily obtained for semiconductor nanoparticles. They showed very beautiful spectral slopes for the regions, from 2 to 5 according to the well known dependence of the fluorescence with the number of photons that excites the material, as it was expected. Also, they used Z-scan technique to calibrate their own 2PA cross-section that will be one the main values to determine the other higher multiphoton absorption cross-sections.

In this point, the authors (presented on SI (Z-scan section)) have obtained a 2PA cross-section on their samples and compared (one of them) to the value reported on the literature. According to the authors, they obtained a value 3 order higher than the one reported {ref 1}, which indicates an "emerging controversy over the contrasting nonlinear two-photon..." , also they justified it by samples diversity. This point is not enough clear. The value, already 3 orders higher, for their case, is used in a phenomenological equation to calculate the other multi-photon high order values.

In my opinion, this method carries to many uncertainties, which can be one of the causes behind the highest values in 5PA.

First, the MEPL strength is a number that depends on a larger number of experimental parameters. For example:

The intensity of the laser beam is one of the uncertainties.

Are the pulse-width the same for all wavelengths? Is it really Gaussian and for all wavelength? The authors must verify these points? Do some autocorrelation to verify the pulse-width and so on.

Another point is the distance of the sample to the focal point, 3.5 cm. At this point, the beam waist will be different for each individual wavelength, which gives a factor on the intensity determination. Thus the authors checked this parameter? It is a crucial point.

Why the author did not performed Z-Scan for the other spectral region, mainly for 2100 nm? Or at list on the 3PA and 4PA region. These regions are at list easily to measure multi-photon absorption with Z-Scan technique, as presented in some references cited on this manuscript. It will help to compare the MEPL method and turns the results stronger by two different experimental procedures in a widely spectral region.

After check these points and make clear why it is 9 orders of magnitude higher, the authors should send this manuscript for a more specific journal on the photochemical and photophysical field. The giant 5 photon absorption cross-section is not enough information to justify a publication in Nature Communications. The sample is already know and multiphoton absorption as well.

REVIEWERS' COMMENTS:

Reviewer #1 (Remarks to the Author):

The authors have addressed all my concerns in the revised MS. I recommend publishing the manuscript.

Reviewer #2 (Remarks to the Author):

Following the reviewers' suggestions, the authors provided additional evidence or more experimental details / figures to address the reviews concerns and strengthen their conclusions. The article, in its present form, is much improved. I am very satisfied with the revised version of the manuscript, thus suggest the acceptance of the manuscript without changes.

Reviewer #3 (Remarks to the Author):

After all questions be well answered by the authors, I decided that this article, in my opinion, can be accepted for publication on Nature Communications.
The three-photon results measured by the authors made the work stronger, show that both techniques, at least for two and three photons are in excellent agreement. All questions were answered accordingly with I was expecting. Also the authors show more results than I had asked, with contributed to solve some doubts that I still had.

Reviewer's comments to the author

Reviewer 1

Prof. Sum and colleagues report high-order nonlinear optical response in the emergent material of perovskite semiconductor nanocrystals by showing efficient PL emission induced by multi-photon absorption in the core-shell halide nanocrystals. As the first experimental evidence of efficient 5-photon process in semiconductor nanocrystals, the work is timely and interesting for the community studying NLO in semiconductor nanomaterials. The conclusions are well supported. Thus, I recommend publishing the paper after addressing some technical concerns summarized below.

Question 1) The authors have emphasized multi-photon imaging as an important potential application for their findings. However, halide perovskite nanocrystals are sensitive to moisture. It seems the material is intrinsically instable in water where multiphoton imaging is conducted. Can the authors discuss on the potential approach to tackle this technical challenge?

Response 1)

We would like to thank the reviewer for the constructive comments. We agree with the reviewer that the moisture sensitivity of halide perovskite nanocrystals could be an issue for potential applications in multi-photon imaging. We have carefully considered this limitation and would like to propose the following three possible encapsulation approaches to circumvent this technical challenge:

(i) Encapsulation with an inert SiO_x/SiO₂ shell/matrix. Although the intrinsic stability of halide perovskites is low in polar solvents such as water, we envisage the use of SiO_x/SiO₂ either as a shell or as a medium to embed several nanocrystals (NCs) to enhance the stability as well as reduce the cytotoxicity from the Pb²⁺ ion. This approach was successfully applied to CdSe (Selvan, S. T. *et al. Adv. Mater.* **17**, 1620-1625, (2005)) core only NCs, and CdSe/ZnS (Gerion, D. *et al. J. Phys. Chem. B* **105**, 8861-8871, (2001); Zhang, T. *et al. Nano Lett.* **6**, 800-808, (2006)), CdSe/CdS (Fan, H. *et al. Nano Lett.* **5**, 645-648, (2005)), CdSe/CdS/ZnS (Jun, S. *et al. ACS Nano* **7**, 1472-1477, (2013)) core-shell NCs with highly encouraging results. In the context of halide perovskite NCs, Huang *et al. (J. Am. Chem. Soc.* **138**, 5749-5752, (2016)) embedded MAPbBr₃ NCs in a SiO_x matrix (~150 nm and 470 nm in diameter) by a 'waterless' sol-gel method using tetramethylorthosilicate (TMOS) as Si-source. The low concentration of water present in analytical grade toluene was successfully used to hydrolyze the TMOS precursor, thus avoiding the addition of extra water as a catalyst to drive the hydrolysis and condensation reactions. This provides a crucial proof-of-concept on the viability of the approach with encapsulated NCs showing increased photostability at 60-80 %RH, without loss of photoluminescence quantum yield (PLQY), although more work is required to achieve finer control of SiO_x/SiO₂ encapsulation for each perovskite NC.

(ii) Encapsulation with capping ligands. In another article, Luo *et al. (Angew. Chem. Int. Ed.* **55**, 8864-8868, (2016)) used 3-aminopropyltriethoxysilane (APTES) as a branched capping ligand to replace the conventional straight-chained ligands such as *n*-octylamine. In addition to an increased control over the NC size (distribution) and improved surface passivation, the APTES-coated MAPbBr₃ NCs show an increased stability in 2-propanol (*i.e.*, 95% retention of PLQY after 2.5 hours), compared

to straight-chained ligands. Since the amino-functionality of APTES successfully passivates the NC surface, the silane group is available for additional sol-gel reactions to further increase the NC stability.

(iii) Encapsulation with solid lipid structures. In a more recent report by Gomez et al. (Gomez, L. *et al. Nanoscale* **9**, 631-636, (2017)), solid lipid structures were used to encapsulate CsPbBr₃ NCs, resulting in a stability up to 2 months in water, albeit at low PLQY (around 11%) due to the initial water degradation during the formation of the solid lipid structures. If this initial degradation can be circumvented, this approach may result in water-stable NCs at high PLQY.

The abovementioned methodologies are highly promising approaches that will address the potential stability and cytotoxicity issues of halide perovskite NCs in aqueous media for multi-photon imaging applications.

In view of the referee's comment, we have added the following discussion to the original manuscript on page 16, paragraph 1 of the revised manuscript:

“The outstanding higher order nonlinear optical properties of the lead bromide perovskite colloidal NCs (particularly the highly efficient five-photon excited upconversion fluorescence in the multi-dimensional core-shell perovskite NCs (*e.g.*, MAPbBr₃/(OA)₂PbBr₄)) indicate their great potential for developing next generation highly efficient, sensitive multi-photon imaging applications with unmatched imaging depth and resolution. Although the intrinsic low stability of halide perovskite NCs in polar solvents (such as water) and their cytotoxicity from the Pb²⁺ ion could be an issue for the potential applications in multi-photon bio-imaging, such technical challenges could be circumvented through the following three possible encapsulation approaches:

(i) Encapsulation with an SiO_x/SiO₂ inert shell/matrix. Utilizing SiO_x/SiO₂ either as an additional shell or as a medium to embed several NCs will help to enhance the stability as well as reduce the cytotoxicity. Such approach has been successfully applied to traditional inorganic semiconductor NCs⁵⁸⁻⁶² for efficient bio-imaging applications. Very recently, encapsulating lead bromide NCs into a SiO_x spherical matrix (about 150 nm and 470 nm in diameter) have been realized by Huang *et al.*⁶³. This provides a crucial proof-of-concept on the viability of the approach, although more work is required to achieve finer control of SiO_x/SiO₂ encapsulation for each perovskite NC.

(ii) Encapsulation with ligands. Through utilizing 3-aminopropyltriethoxysilane (APTES) as a branched capping ligand to replace the conventional straight-chained ligands, Luo *et al.*⁶⁴ achieved enhanced stability in the APTES-coated MAPbBr₃ NCs. Such APTES-coated MAPbBr₃ NCs show an increased stability in 2-propanol. Furthermore, the available silane group allows for additional sol-gel reactions which may further increase the stability of the NCs.

(iii) Encapsulation with solid lipid structures. More recently, Gomez *et al.*⁶⁵ reported the application of solid lipid structures to encapsulate CsPbBr₃ NCs that were stable up to 2 months in water, albeit at a low PLQY (around 11%) likely due to initial water degradation. Further work is needed to circumvent the initial degradation and realize water-stable NCs at high PLQY.

These examples clearly show that the potential stability and cytotoxicity issues of halide perovskite NCs in aqueous media for multi-photon imaging applications could in fact be overcome through a judicious choice of the encapsulation approach and careful optimization.

New references 58-65 were added to the manuscript:

58. Selvan, S. T., Tan, T. T. & Ying, J. Y. Robust, Non-Cytotoxic, Silica-Coated CdSe Quantum Dots with Efficient Photoluminescence. *Adv. Mater.* **17**, 1620-1625, (2005).
59. Gerion, D. *et al.* Synthesis and Properties of Biocompatible Water-Soluble Silica-Coated CdSe/ZnS Semiconductor Quantum Dots. *J. Phys. Chem. B* **105**, 8861-8871, (2001).
60. Zhang, T. *et al.* Cellular Effect of High Doses of Silica-Coated Quantum Dot Profiled with High Throughput Gene Expression Analysis and High Content Cellomics Measurements. *Nano Letters* **6**, 800-808, (2006).
61. Fan, H. *et al.* Surfactant-Assisted Synthesis of Water-Soluble and Biocompatible Semiconductor Quantum Dot Micelles. *Nano Letters* **5**, 645-648, (2005).
62. Jun, S., Lee, J. & Jang, E. Highly Luminescent and Photostable Quantum Dot–Silica Monolith and Its Application to Light-Emitting Diodes. *ACS Nano* **7**, 1472-1477, (2013).
63. Huang, S. *et al.* Enhancing the Stability of CH₃NH₃PbBr₃ Quantum Dots by Embedding in Silica Spheres Derived from Tetramethyl Orthosilicate in “Waterless” Toluene. *Journal of the American Chemical Society* **138**, 5749-5752, (2016).
64. Luo, B. *et al.* Organolead Halide Perovskite Nanocrystals: Branched Capping Ligands Control Crystal Size and Stability. *Angewandte Chemie International Edition* **55**, 8864-8868, (2016).
65. Gomez, L., de Weerd, C., Hueso, J. L. & Gregorkiewicz, T. Color-stable water-dispersed cesium lead halide perovskite nanocrystals. *Nanoscale* **9**, 631-636, (2017).

Question 2) The slope of the power dependence of PL on the incident power has been taken as a metric to distinguish the order of multi-photon processes with excitation of different wavelengths (Figure 2a). I guess the ranges of the excitation power are different for different wavelengths. It is likely that the higher-order processes dominate if the power is sufficiently high even with higher photon energy. The authors should include the information of power ranges for each wavelengths used in these measurements.

Response 2)

We thank the reviewer for the constructive feedback. Yes, the ranges of the excitation power/fluence are different for different wavelengths. We agree that the excitation fluence ranges for different orders of multi-photon processes are not clearly conveyed through the inset plots in the original figures 1b and S6a-c showing the excitation fluence dependence of spectrally integrated upconversion PL intensity. In these figures, larger upconversion PL intensities were obtained at lower excitation fluences in the lower order multi-photon processes compared to the higher ones. Together with the slopes of excitation fluence dependences of integrated upconversion PL intensity, one can rule out the presence of higher-order processes at higher excitation photon energies. To present such information more clearly, we have revised the related descriptions of the multi-photon processes to include the excitation fluence ranges:

1. In original manuscript, lines 98-100: “Figure 1b shows the 5P-excited (i.e., at 2100 nm wavelength) excitonic PL emission (centred around 520 nm) from these samples and their quintic excitation fluence dependence (inset).”

was changed to

“Figure 1b shows the 5P-excited (i.e., at 2100 nm wavelength) excitonic PL emission (centred around 520 nm) from these samples with excitation fluence range about 3.30 – 5.61 mJ/cm² (inset)

and their quintic excitation fluence dependence (inset).” on page 6, paragraph 1, lines 4-8 of the revised manuscript

2. In original manuscript, lines 115-119: “*Similar to figure 1b, MEPL spectra under 2, 3 and 4-photon excitation for the NCs at 800, 1200, 1600 nm, respectively, are shown in figure S6a-c; and illustrated schematically in figure S6d. Figures S6a-c insets clearly show the nearly quadratic, cubic and quartic dependences of the spectrally integrated PL intensity on excitation fluence for the 2, 3 and 4-photon processes, respectively.*”

was changed to:

“Similar to figure 1b, MEPL spectra under 2, 3 and 4-photon excitation for the NCs at 800, 1200, 1600 nm with excitation fluence ranges about 0.33 – 1.16 mJ/cm², about 0.66 – 1.82 mJ/cm², about 1.16 – 2.31 mJ/cm² (i.e., higher upconversion PL intensities in the lower-order multi-photon processes even at lower excitation fluence), respectively, are shown in figure S11a-c; and illustrated schematically in figure S11d. Figures S11a-c insets clearly show the nearly quadratic, cubic and quartic dependences of the spectrally integrated PL intensity on excitation fluence for the 2, 3 and 4-photon processes with excitation fluence ranges about 0.33 – 1.16 mJ/cm², about 0.66 – 1.82 mJ/cm², about 1.16 – 2.31 mJ/cm², respectively.” on page 7, paragraph 1, lines 5-12 of the revised manuscript.

3. In original SI, lines 199-205: “*The IPA spectra of the NCs in figure S1a-c reveal absorption maxima at ~ 505 nm & ~ 510 nm, and negligible absorbance in wavelengths longer than ~ 530 nm. Consequently, the PL spectra from the NCs acquired at 800, 1200, 1600, and 2100 nm excitation illustrated in figure S6a-c and figure 1b in main text correspond to the two-, three-, four- and five-photon absorption processes. Moreover, the nearly quadratic, cubic, biquadratic and quintic excitation fluence dependence of the spectrally integrated PL intensity shown in the insets of figures S6a-c and figure 1b further justify that MPA is responsible for the fluorescence excitation at infrared wavelengths in these NCs.*”

was changed to

“The IPA spectra of the NCs in figure S1a-c reveal absorption maxima at ~ 505 nm & ~ 510 nm, and negligible absorbance in wavelengths longer than ~ 530 nm. Consequently, the PL spectra from the NCs acquired at 800, 1200, 1600, and 2100 nm excitation illustrated in figure S11a-c and figure 1b in main text correspond to the two-, three-, four- and five-photon absorption processes (excitation fluence ranges ~ 0.33 – 1.16 mJ/cm², ~ 0.66 – 1.82 mJ/cm², ~ 1.16 – 2.31 mJ/cm², ~ 3.30 – 5.61 mJ/cm²; larger upconversion PL intensity in lower-order multi-photon processes even at lower excitation fluence). Moreover, the nearly quadratic, cubic, biquadratic and quintic excitation fluence dependence of the spectrally integrated PL intensity shown in the insets of figures S11a-c and figure 1b with excitation fluence ranges ~ 0.33 – 1.16 mJ/cm², ~ 0.66 – 1.82 mJ/cm², ~ 1.16 – 2.31 mJ/cm², ~ 3.30 – 5.61 mJ/cm² further justify that MPA is responsible for the fluorescence excitation at infrared wavelengths in these NCs.” on page 27, paragraph 1, lines 1-10 in the revised SI

Question 3) In the discussion part, the authors have included in-depth discussion on the enhancement of multi-photon process in core-shell nanocrystals with respect to that to core-only nanocrystals, which is good. However, the intrinsic mechanism why perovskite semiconductor nanocrystals have a dramatically enhanced MPA has not been explained. Moreover, the authors note the possible involvement of quantum confinement in nanocrystals. Is it possible to show a more detailed dependence of MPA to explain whether the excitonic effect relevant to the quantum confinement effect is critical?

Response 3)

We would like to thank the reviewer for the constructive comments. The superior MPA of the lead bromide perovskite NCs compared to the traditional semiconductor NCs result from both the intrinsic strong MPA of lead bromide perovskites (Wang, Y. *et al. Nano Lett.* **16**, 448-453 (2016); Xu, Y. *et al. J. Am. Chem. Soc.* **138**, 3761-3768 (2016)) and the quantum confinement effect (Wang, Y. *et al. Nano Lett.* **16**, 448-453 (2016); Xu, Y. *et al. J. Am. Chem. Soc.* **138**, 3761-3768 (2016); Larson, D. R. *et al. Science* **300**, 1434-1436 (2003)). Quantum confinement has been demonstrated to greatly enhance the MPA of traditional semiconductor NCs (Larson, D. R. *et al. Science* **300**, 1434-1436 (2003)). The intrinsic strong MPA of lead bromide perovskites and quantum confinement effect was found to give rise to the giant 2PA in CsPbBr₃ NCs (Wang, Y. *et al. Nano Lett.* **16**, 448-453 (2016); Xu, Y. *et al. J. Am. Chem. Soc.* **138**, 3761-3768 (2016)). Although the sizes of the MAPbBr₃ (~ 8-9 nm), MAPbBr₃/(OA)₂PbBr₄ (~ 9-10 nm) and CsPbBr₃ NCs (~ 9 nm) are larger than their corresponding Bohr diameters ($d_B \sim 4$ nm for MAPbBr₃ (Tanaka, K. *et al. Solid State Commun.* **127**, 619-623 (2003)) and $d_B \sim 7$ nm for CsPbBr₃ (Protesescu *et al. Nano Lett.* **15**, 3692-3696 (2016)), respectively), the clear blue shifts of ~15 nm and ~45 nm in the 1PA peaks compared to their bulk counterparts (Tyagi, P. *et al. J. Phys. Chem. Lett.* **6**, 1911-1916 (2015); Zhang, M. *et al. Chem. Commun.* **50**, 11727-11730 (2014); Stoumpos, C. C. *et al. Cryst. Growth Des.* **13**, 2722-2727 (2013)) provides clear evidence of the weak quantum confinement in these perovskite NCs (as in original SI, lines 23-37). Hence it is fair to classify our perovskite NCs to be in the weak confinement regime.

The larger blue shift in CsPbBr₃ NC stems from its relatively stronger quantum confinement due to its larger Bohr diameter relative to the sample size (*i.e.*, $d_B \sim 7$ nm vs 9 nm sample size) compared to MAPbBr₃ NC ($d_B \sim 4$ nm vs 8-9 nm sample size). In the straightforward comparison between the MPA of MAPbBr₃ and CsPbBr₃ NCs with similar sample size of ~9 nm, the effect of quantum confinement on the MPA of lead bromide perovskite NCs manifests itself. In fact, the CsPbBr₃ NCs with relatively stronger quantum confinement demonstrate ~3-4 times larger MPA than MAPbBr₃ NCs (as discussed in the main text and shown in insets of figure 1d and 2b-d in original manuscript).

In the original manuscript, lines 203-206, we have mentioned: “Given that our MAPbBr₃ NCs are in the weak confinement regime ($d_B \sim 4$ nm \ll 8-9 nm diameter), it is possible to enhance these σ_n values even further through even smaller strongly-confined MAPbBr₃ NCs and their core-shell counterparts (*i.e.*, NCs’ diameter ~ 4nm).” Detailed investigations of the quantum confinement effect on the MPA of lead bromide perovskite NCs would require the synthesis of very small size & strongly quantum confined MAPbBr₃ (< 4 nm) and CsPbBr₃ NCs (< 7 nm) with high crystalline quality and uniform size distribution. However, this is presently still very challenging for perovskite NCs due to various reasons (as outlined below).

The all-inorganic CsPbBr₃ NCs are prepared via the hot-injection method at temperatures between 140-180 °C. The NC size is determined by the reaction temperature, where larger NCs are obtained at higher temperature. However, below 140 °C, nanoplatelets (not NCs) are formed instead (Bekenstein *et al. J. Am. Chem. Soc.* **137**, 16008-16011 (2015) and Akkerman *et al. J. Am. Chem. Soc.* **138**, 1010-1016 (2015)). Therefore, it is not feasible to synthesize CsPbBr₃ NCs with size smaller than Bohr radius by simply lowering the temperature, CsPbBr₃ nanoplatelets will gradually form at reduced temperature.

For MAPbBr₃ NCs, Teunis *et al. (Nanoscale* **9**, 17433-17439, (2016)) used diaminododecane (DDAD) and hexylamine (HA) as ligands to control the sizes of NCs during the ligand-assisted reprecipitation method. The reaction yielded particles with average size ~1.5 nm, which exhibited white-light emission caused by the surface defects. This indicates that the concentration of surface defect states dominate when reducing the NC size below the Bohr diameter, which will unambiguously and negatively influence the MPA. Although Huang *et al. (Adv. Sci.* **2**: 1500194, (2015)) have applied both the very high speed centrifugation (14,500 rpm, *i.e.*, size exclusion) and controlling the reaction temperature to synthesize MAPbBr₃ NCs with average size below the Bohr diameter, the reaction yield through centrifugation at such high speeds is extremely low and not feasible for higher-order MPA study. In addition, even though using high speed centrifugation (14,500 rpm, *i.e.*, size exclusion) can obtain MAPbBr₃ NCs with small sizes, the size distribution is uncontrollable since all NCs with sizes below certain value will be included in the resulting product. Hence, at current stage, despite MAPbBr₃ NCs with small size can be obtained, various problems associated with high density surface defects, low sample crystalline quality, low reaction yield and non-uniform size distribution need to be further addressed.

To conclude, at current stage, it is extremely challenging to synthesize small MAPbBr₃ and CsPbBr₃ NCs with high crystalline quality, low surface defects, uniform size distribution, and relatively high reaction yield, which are needed to conduct detailed investigations of the quantum confinement effect on the MPA of lead bromide perovskite NCs. Presently, there have been no reports on the synthesis of small-sized core-shell MAPbBr₃/(OA)₂PbBr₄ NCs. Significant advances in perovskite NCs synthesis are needed before this question on the detailed dependence of MPA on the quantum confinement effect could be answered.

In view of the reviewer's comment, we have revised the related discussion in the original manuscript:

In the original manuscript, lines 203-206 “Given that our MAPbBr₃ NCs are in the weak confinement regime ($d_B \sim 4 \text{ nm} \ll 8\text{-}9 \text{ nm}$ diameter), it is possible to enhance these σ_n values even further through even smaller strongly-confined MAPbBr₃ NCs and their core-shell counterparts (*i.e.*, NCs' diameter $\sim 4\text{nm}$)”

was changed to:

“Given that **both** our MAPbBr₃ and CsPbBr₃ NCs are in the weak confinement regime ($d_B \sim 4 \text{ nm} \ll 8\text{-}9 \text{ nm}$ diameter **for MAPbBr₃ and $d_B \sim 7 \text{ nm} < 9 \text{ nm}$ diameter for CsPbBr₃**), it is possible to enhance these σ_n values even further through even smaller strongly-confined MAPbBr₃ NCs, their core-shell counterparts (*i.e.*, NCs' diameter $< 4\text{nm}$) **and CsPbBr₃ NCs (*i.e.*, NCs' diameter $< 7 \text{ nm}$). However,**

at current stage, it is extremely challenging to synthesize small MAPbBr₃^{50,51} and CsPbBr₃ NCs^{52,53} with high crystalline quality, low surface defects, uniform size distribution, and relatively high reaction yield, which are needed to conduct detailed investigations of the quantum confinement effect on the MPA of lead bromide perovskite NCs. The all-inorganic CsPbBr₃ NCs were prepared via the hot-injection method⁴⁰ at temperatures between 140-180 °C. The NC size is determined by the reaction temperature, where larger NCs are obtained at higher temperature. However, below 140 °C, nanoplatelets (not NCs) are formed instead^{52,53}. Moreover, at current stage, despite MAPbBr₃ NCs with small size can be obtained^{50,51}, various problems associated with high density surface defects, low sample crystalline quality, low reaction yield and non-uniform size distribution need to be further addressed^{50,51}. Presently, there have been no reports on the synthesis of small-sized core-shell MAPbBr₃/(OA)₂PbBr₄ NCs. Therefore, significant advances in perovskite NCs synthesis are needed before this question on the detailed dependence of MPA on the quantum confinement effect could be answered.” on page 13, paragraph 1, line 2-18 of the revised manuscript.

New references 50-53 **were added** to the manuscript:

50. Bekenstein, Y., Koscher, B. A., Eaton, S. W., Yang, P. & Alivisatos, A. P. Highly Luminescent Colloidal Nanoplates of Perovskite Cesium Lead Halide and Their Oriented Assemblies. *J. Am. Chem. Soc.* **137**, 16008-16011, (2015).
51. Akkerman, Q. A. et al. Solution Synthesis Approach to Colloidal Cesium Lead Halide Perovskite Nanoplatelets with Monolayer-Level Thickness Control. *J. Am. Chem. Soc.* **138**, 1010-1016, (2016).
52. Teunis, M. B., Lawrence, K. N., Dutta, P., Siegel, A. P. & Sardar, R. Pure white-light emitting ultrasmall organic-inorganic hybrid perovskite nanoclusters. *Nanoscale* **8**, 17433-17439, (2016).
53. Huang, H., Susha, A. S., Kershaw, S. V., Hung, T. F. & Rogach, A. L. Control of Emission Color of High Quantum Yield CH₃NH₃PbBr₃ Perovskite Quantum Dots by Precipitation Temperature. *Adv. Sci.*, **2**: 1500194, (2015).

Question 4) In Figure 1(c) & Figure S6, the authors have schematically illustrated the transition induced by MPA in momentum space. In general, the momentum of a photon is negligible in compared to that of an electron. The processes of efficient MPA and PL emission here are likely to be direct transitions without involvement of any phonons. I would suggest the authors to draw these transitions vertically to avoid potential misunderstandings.

Response 4)

We would like to thank the reviewer for the constructive feedback. Yes, no phonon is involved in the processes of efficient MPA and resulted upconversion PL emission in the manuscript. It is indeed confusing and ambiguous to depict the multiphoton absorption processes schematically with a slight tilt in our figures. To present the results more clearly and to avoid potential misunderstandings, we have revised figure 1(c) and figure S6 accordingly.

Original figure 1,

Figure | 1 Type I core-shell multi-dimensional perovskite NCs and their ultra-large five-photon action cross-sections ($\eta\sigma_5$). (a) Schematic illustrating the core-shell multi-dimensional perovskite NCs with 3D MAPbBr₃ as core and 2D (OA)₂PbBr₄ as shell, and their type-I energy level alignment; (b) 5PPL spectra from core-only MAPbBr₃ NCs (~2.0 μM in toluene), core-shell MAPbBr₃/(OA)₂PbBr₄ NCs (~2.1 μM in toluene) and CsPbBr₃ NCs (~1.0 μM in toluene), with femtosecond laser excitation at 2100 nm. Inset shows the quintic dependence on the excitation fluence of the spectrally integrated PL intensity; (c) Schematic illustrating the 5PPL process in perovskite NCs; (d) five-photon action cross-section ($\eta\sigma_5$) spectra of the MAPbBr₃, MAPbBr₃/(OA)₂PbBr₄ and CsPbBr₃ NCs. Error bars indicate experimental uncertainty of $\pm 15\%$.

was changed to:

Figure | 1 Type I core-shell multi-dimensional perovskite NCs and their ultra-large five-photon action cross-sections ($\eta\sigma_5$). (a) Schematic illustrating the core-shell multi-dimensional perovskite NCs with 3D MAPbBr₃ as core and 2D (OA)₂PbBr₄ as shell, and their type-I energy level alignment; (b) 5PPL spectra from core-only MAPbBr₃ NCs (~2.0 μM in toluene), core-shell MAPbBr₃/(OA)₂PbBr₄ NCs (~2.1 μM in toluene) and CsPbBr₃ NCs (~1.0 μM in toluene), with femtosecond laser excitation at 2100 nm. Inset shows the quintic dependence on the excitation fluence of the spectrally integrated PL intensity; (c) Schematic illustrating the 5PPL process in perovskite NCs; (d) five-photon action cross-section ($\eta\sigma_5$) spectra of the MAPbBr₃, MAPbBr₃/(OA)₂PbBr₄ and CsPbBr₃ NCs. Error bars indicate experimental uncertainty of $\pm 15\%$.

Original figure S6,

Figure S6 | Two, three, and four-photon excited upconversion PL spectra from the toluene solution of MAPbBr₃ NCs (~ 2.0 μ M), MAPbBr₃/(OA)₂PbBr₄ NCs (~ 2.1 μ M) and CsPbBr₃ NCs (~ 1.0 μ M) excited by fs laser beam at 800, 1200, and 1600 nm. Insets indicate their excitation fluence dependence by the log-log plots of spectrally integrated PL intensity vs excitation laser fluences. (a-c) Two-, three- and four-photon-absorption excited PL spectra from the perovskite NCs at 800, 1200, 1600 nm, respectively. Insets are their corresponding quadratic, cubic and quartic dependence on the excitation fluence; (d) Schematic illustrating the processes of 2PPL, 3PPL and 4PPL in the perovskite NCs corresponding to (a - c).

was changed to:

Figure S11 | Two, three, and four-photon excited upconversion PL spectra from the toluene solution of MAPbBr₃ NCs (~ 2.0 μM), MAPbBr₃/(OA)₂PbBr₄ NCs (~ 2.1 μM) and CsPbBr₃ NCs (~ 1.0 μM) excited by fs laser beam at 800, 1200, and 1600 nm. Insets indicate their excitation fluence dependence by the log-log plots of spectrally integrated PL intensity vs excitation laser fluences. (a-c) Two-, three- and four-photon-absorption excited PL spectra from the perovskite NCs at 800, 1200, 1600 nm, respectively. Insets are their corresponding quadratic, cubic and quartic dependence on the excitation fluence; (d) Schematic illustrating the processes of 2PPL, 3PPL and 4PPL in the perovskite NCs corresponding to (a - c).

Reviewer 2

Chen et al. reported the high-order nonlinear multiphoton upconverting luminescence (UCL) in the family of halide perovskite colloidal semiconductor nanocrystals (NCs). Specifically, large 5-photon action cross-sections ($\eta\sigma^5$) that is at least 9-orders larger than state-of-the-art specially-designed organic molecules was first demonstrated. It was revealed that the multi-dimensional core-shell nanostructure design contributed significantly to such large value of $\eta\sigma^5$. The results are convincing and very important in the fields of both perovskite NCs and multiphoton UCL. The manuscript is well organized and scholarly written in a very high quality. Therefore, I recommend the publication of this manuscript in Nat. Commun. after the following minor revisions:

Question 1. The authors stated that higher-order nonlinear UCL are more favorable for bioimaging (e.g., high spatial confinement, long penetration depth, and low biological damage). However, the low efficiency of the high-order nonlinear UCL may limit its practical bioapplications. Meanwhile, the Pb²⁺ ion in the matrix of the perovskite NCs is toxic, and the halide perovskite NCs have a poor stability and are incompatible with biosystems in view of their hydrophobic nature. The authors are suggested to address this concern with more discussion or perspectives.

Response 1

We thank the reviewer for the constructive comments and feedback. The relatively low efficiency of the high-order nonlinear UCL could possibly hinder its practical bio-applications. However, we would like to point out that the higher-order nonlinear UCL, although lower in efficiency, could still be demonstrated with our perovskite NCs. This is in stark contrast for conventional inorganic NCs, where there have been no reports for 4- or 5-photon UCL. This is precisely the main reason why most of the reported multi-photon fluorescence microscopies were based on lower-order two-photon excited upconversion fluorescence (Denk, W. *et al. Science* **248**, 73-76 (1990); Larson, D. R. *et al. Science* **300**, 1434-1436 (2003)). Another factor to consider in multi-photon fluorescence microscopy is the advances in detector technology and improvements to the detection efficiency, which would mitigate the lower efficiency of the higher order UCL. For instance, very recently, three-photon fluorescence microscopy of subcortical structures within an intact mouse brain was reported utilizing the fluorescent dyes and fluorescent proteins excited at around 1700 nm (Horton, N. G. *et al. Nat. Photon.* **7**, 205-209 (2013)).

The low efficiency of high-order nonlinear UCL will also require higher excitation fluence/power, which may cause tissue heating, cell damage or even the damage to the multi-photon fluorescent label. Two parameters of the excitation laser beams are essential for this consideration, the excitation fluences (peak excitation intensities) and the average excitation intensities. The high average excitation intensities may cause the samples/targets heating, which is usually a problem when utilizing high-repetition-rate and low-pulse-energy excitation laser source or continuous laser source (Horton, N. G. *et al. Nat. Photon.* **7**, 205-209 (2013), Kobat, D. *et al. Opt. Express* **17**, 13354–13364 (2009)). On the other hand, the heating effect seldom happens when utilizing low-repetition-rate and high-pulse-energy excitation laser source (Horton, N. G. *et al. Nat. Photon.* **7**, 205-209 (2013), Kobat, D. *et al. Opt. Express* **17**, 13354–13364 (2009)). In our case, low-repetition-rate (1kHz) and high-pulse-energy excitation laser beam was applied. In addition, the low average excitation intensities ($(I_{\text{avg}})_{\text{max}} \sim 5.61 \text{ W/cm}^2$) was utilized to achieve relative large five-photon excited upconversion PL emission, which is visible to eye (figure 1b in the original manuscript and figure S7 in the original SI). Such average excitation intensity is much lower than the one utilized in three-photon fluorescence microscopy ($\geq 2.36 \text{ kW/cm}^2$) where higher repetition rate excitation laser source (1MHz) was applied

(Horton, N. G. *et al. Nat. Photon.* **7**, 205-209 (2013)). Hence, the heating effect is not expected to be an issue for perovskite NCs for bio-imaging applications with higher order UCL.

Another factor to consider is the structural damage of the samples/targets due to the large peak excitation intensities used. In our case, the maximum peak intensity utilized to excite the relatively large five-photon upconversion emission (as in figure 1b in the original manuscript and figure S7 in the original SI) is about 90 GW/cm². Such peak excitation intensity is smaller than that applied in the three-photon fluorescence microscopy (about 200 GW/cm²) (Horton, N. G. *et al. Nat. Photon.* **7**, 205-209 (2013)), and it will not induce structural damage to the samples/targets. Hence, the structural damage of the samples/targets induced by the high peak excitation intensities may not be an issue for the potential applications of the higher-order UCL (three-, four- and five-photon UCL) from perovskite NCs in bio-imaging. The heating effect and structural damage could in fact be mitigated even further in multi-photon fluorescent labels with very large action cross-sections as the required intensities would be even lower. This is indeed the case for our perovskite NCs, which has very large multi-photon action cross-sections.

In summary, we feel that the relatively low efficiency of the higher-order nonlinear UCL of perovskite NCs would not be an obstacle to practical bio-applications.

The reviewer's second comment on the toxicity and stability of the perovskite NCs is similar to the concern raised by Reviewer 1 (Question 1). These issues can be overcome by several encapsulation approaches: (i) Encapsulation with an inert SiO_x/SiO₂ shell/matrix; (ii) Encapsulation with ligands; and (iii) Encapsulation with solid lipid structures. Please refer to the detailed response to Reviewer 1 Question 1 on page 1 of this response letter.

Question 2. For the potential bioapplication of UCL, some most relevant papers might be included in the background Introduction.

Response 2

We thank the reviewer for the constructive comments and feedback. We have added the most relevant papers on the potential bioapplication of UCL at the following locations of our revised manuscript:

In the original manuscript, lines 38-43: “*Such frequency-upconverted fluorescence possesses many attractive merits over linearly excited fluorescence (e.g., high spatial confinement, long penetration depth, and low biological damage)¹⁻⁴ due to the longer infrared excitation wavelengths. Two-photon/three-photon absorption (2PA/3PA) applications have found widespread use in three-dimensional (3D) biomedical imaging¹⁻⁴, optical power limiting⁵, sensing⁶ and 3D optical data storage⁷ etc.*” **was changed to** “Such frequency-upconverted fluorescence possesses many attractive merits over linearly excited fluorescence (e.g., high spatial confinement, long penetration depth, and low biological damage)¹⁻⁷ due to the longer infrared excitation wavelengths. Two-photon/three-photon absorption (2PA/3PA) applications have found widespread use in three-dimensional (3D) biomedical imaging¹⁻⁷, optical power limiting⁸, sensing⁹ and 3D optical data storage¹⁰ etc.” on page 3, paragraph 1, line 3 of the revised manuscript.

New references 5-7 **were added** to the manuscript:

5. Yong, K.-T. *et al.* Quantum Rod Bioconjugates as Targeted Probes for Confocal and Two-Photon Fluorescence Imaging of Cancer Cells. *Nano Lett.* **7**, 761-765, (2007).
6. Zhang, C. *et al.* Multiphoton absorption induced amplified spontaneous emission from biocatalyst-synthesized ZnO nanorods. *Appl. Phys. Lett.* **92**, 233116, (2008).
7. Horton, N. G. *et al.* In vivo three-photon microscopy of subcortical structures within an intact mouse brain. *Nat. Photon.* **7**, 205-209, (2013).

Reviewer 3

According to authors, the manuscript reports for the first time five-photon absorption on NC's "5-P-excited upconversion ... has yet to be demonstrated." Also, the authors reported that the value is about 9 orders higher than the best organic molecule reported on the literature. The main subject of the manuscript is about the higher five photon absorption measured and determined by the authors. As the authors said, five photons, or multi-photon absorption were already studied in a large number of organic and inorganic materials, as can be seen on the references cited on the manuscript. But the fact of have an enhancement of 9 order compared to the best organic molecules motivates NC, showed on this manuscript, to be used on multiphoton fluorescence microscopy.

Response to general comments:

We would like to thank the reviewer for his precious time and valuable comments. We agree with the reviewer that lower-order multi-photon absorption processes (*i.e.*, two- and three-photon absorption) has been well-studied in a large number of organic and inorganic materials, in particular the work by the pioneers in the field *e.g.*, He, G. S.; Zheng, Q.; Prasad, P. N.; Larson, D. R.; Albota, M.; Michalet, X.; Resch-Genger, U.; Xu, C.; Webb, W.; Van Stryland, E. W. *et al*). We have cited all these background work in our manuscript. For example in the original manuscript, lines 44-49:

*"For such applications, materials with large multi-photon absorption cross-sections (σ_n) and high photoluminescence quantum yield (PLQY, η) are required, hence the parameter known as action cross-section (*i.e.*, $\eta \times \sigma_n$). Suitable media include: conventional inorganic semiconductor nanocrystals (NCs)⁹⁻¹¹, organic chromophores¹²⁻¹⁴, polymers^{15,16}, metal complexes¹⁷ and biomolecules¹⁸. Amongst them, the former exhibit exceptional performances with relatively large 2- and 3-photon action cross-sections ($(\eta\sigma_2)_{max} \sim 10^4$ GM and $(\eta\sigma_3)_{max} \sim 10^{-76} \text{cm}^6 \text{s}^2 / \text{photon}^2$)^{8-14,19,20}."*

However, to the best of our knowledge, there are only two reports to date on five-photon absorption in a specially-designed organic molecule (Zheng, Q. *et al. Nat. Photon.* **7**, 234-239 (2013)) and novel ladder-type oligomers (Fan, H. H. *et.al., J. Am. Chem. Soc.* **134**, 7297-7300 (2012)). Quantitative characterization of the five-photon absorption process was only performed for the specially-designed organic molecule in the *Nature Photonics* paper; whereas the second manuscript focus mainly on reporting the large two- and three-photon absorption properties in the novel ladder-type oligomers. Furthermore, to date, five-photon excited upconversion fluorescence from semiconductor NCs has yet to be demonstrated.

Hence, we respectfully disagree with the reviewer's comment that: "*As the authors said, five photons, or multi-photon absorption were already studied in a large number of organic and inorganic materials, as can be seen on the references cited on the manuscript.*"

Question 1) I could say that the results could be interesting to the field. However, the main point that I will question is the method used to confirm the high order absorption cross-section measured.

Of course, the authors carefully demonstrated that the fluorescence processes for different wavelengths regions are dominated by different order of multiphoton absorption, easily obtained for semiconductor nanoparticles. They showed very beautiful spectral slopes for the regions, from 2 to 5 according to the well known dependence of the fluorescence with the number of photons that excites the material, as it was expected. Also, they used Z-scan technique to calibrate their own 2PA cross-

section that will be one the main values to determine the other higher multiphoton absorption cross-sections.

In this point, the authors (presented on SI (Z-scan section)) have obtained a 2PA cross-section on their samples and compared (one of them) to the value reported on the literature. According to the authors, they obtained a value 3 order higher than the one reported {ref 1}, which indicates an “emerging controversy over the contrasting nonlinear two-photon...” , also they justified it by samples diversity. This point is not enough clear. The value, already 3 orders higher, for their case, is used in a phenomenological equation to calculate the other multi-photon high order values.

Response 1)

We would like to thank the reviewer for his valuable time and constructive comments. The comparison between our results and the previous reports may not have been clearly conveyed, leading to some misunderstanding: “According to the authors, they obtained a value 3 order higher than the one reported {ref 1}, which indicates an “emerging controversy over the contrasting nonlinear two-photon...” , also they justified it by samples diversity. This point is not enough clear. The value, already 3 orders higher, for their case, is used in a phenomenological equation to calculate the other multi-photon high order values.”

1. In our manuscript, we did not claim that our measured 2PA cross-section values (at 800 nm) for CsPbBr₃ NCs are 3 orders higher than the one reported in {ref 1, i.e., Wang, Y. *et al. Nano Lett.* **2016**, 16, 448-453}. Our measured 2PA cross-section (σ_2) of cubic CsPbBr₃ NCs (size about 9 nm) in toluene by open-aperture Z-scans at 800 nm is about $(2.0 \pm 0.3) \times 10^6$ GM. Such value is comparable to a very recent report by Xu, Y. *et al.* (Xu, Y. *et al. J. Am. Chem. Soc.* **138**, 3761-3768 (2016)), who obtained σ_2 value of cubic CsPbBr₃ NCs (size about 9 nm) at 800 nm about 2.7×10^6 GM utilizing the same open-aperture Z-scan technique. On the other hand, our result is about one order larger than another recent report by Wang, Y. *et al.* (Wang, Y. *et al. Nano Lett.* **16**, 448-453 (2016)), where σ_2 value of cubic CsPbBr₃ NCs (size about 9 nm) at 800 nm about 1.2×10^5 GM measured by the same open-aperture Z-scan technique was acquired. As illustrated in Xu, Y. *et al. J. Am. Chem. Soc.* **138**, 3761-3768 (2016), a plausible origin of the difference in the obtained σ_2 values of the CsPbBr₃ NCs might be the sample diversity utilized in the different measurements. The above discussion was described in the lines 147-152 of original SI:

In the original SI, lines 145-152:

“2PA cross-sections (σ_2) of the MAPbBr₃, MAPbBr₃/(OA)₂PbBr₄ and CsPbBr₃ NCs at 800 nm were estimated to be $\sim (0.8 \pm 0.1) \times 10^6$ GM, $\sim (5.0 \pm 0.8) \times 10^6$ GM and $\sim (2.0 \pm 0.3) \times 10^6$ GM, respectively. The derived σ_2 value of the CsPbBr₃ NCs at 800 nm is close to the measured result in a recent report², indicating that the Z-scan measurements in current work were appropriately conducted. On the other hand, the estimated σ_2 value of the CsPbBr₃ NCs at 800 nm here is larger than another published value¹. A plausible origin of the difference in the obtained σ_2 values might be the sample diversity utilized in the different measurements, as illustrated in Ref. [2].”

where Refs. [1] and [2] are:

1. Wang, Y. *et al.* Nonlinear Absorption and Low-Threshold Multiphoton Pumped Stimulated Emission from All-Inorganic Perovskite Nanocrystals. *Nano Lett.* **16**, 448-453 (2016).

- Xu, Y. *et al.* Two-Photon-Pumped Perovskite Semiconductor Nanocrystal Lasers. *J. Am. Chem. Soc.* **138**, 3761-3768 (2016).

To describe the comparison more clearly, we have now revised the lines 145-152.

In the original SI, lines 145-152 **were changed to:**

“2PA cross-sections (σ_2) of the MAPbBr₃, MAPbBr₃/(OA)₂PbBr₄ and CsPbBr₃ NCs at 800 nm were estimated to be $\sim (0.8 \pm 0.1) \times 10^6$ GM, $\sim (5.0 \pm 0.8) \times 10^6$ GM and $\sim (2.0 \pm 0.3) \times 10^6$ GM, respectively. The derived σ_2 value of the CsPbBr₃ NCs at 800 nm is close to the measured result in a recent report² (Table S2), indicating that the Z-scan measurements in current work were appropriately conducted. On the other hand, the estimated σ_2 value of the CsPbBr₃ NCs at 800 nm here is **about one order** larger than another published value¹ (Table S2). A plausible origin of the difference in the obtained σ_2 values might be the sample diversity utilized in the different measurements, as illustrated in Ref. [2].”

We have also added a new table to the SI (Table S2) to allow clear comparison between our Z-scan measurements at 800nm on CsPbBr₃ NCs with the reported results in the literature, as shown below.

- Next, the “emerging controversy” refers to the contrasting conclusions of 2PA vs Saturable absorption (SA) reported by refs. [Wang, Y. *et al.* *Nano Lett.* **16**, 448-453 (2016); Xu, Y. *et al.* *J. Am. Chem. Soc.* **138**, 3761-3768 (2016)] and ref. [Lu, W.-G. *et al.* *Adv. Optical Mater.*, DOI:10.1002/adom.201600322 (2016)] on the CsPbBr₃ NCs, respectively. Both Wang, Y. *et al.* and Xu, Y. *et al.* report the observation of 2PA in CsPbBr₃ NCs (cubic, size about 9 nm) in toluene through open-aperture Z-scan at 800 nm. In contrast, utilizing the similar open-aperture Z-scan technique, Lu, W.-G. *et al.* reported SA in both about 5 nm size spherical CsPbBr₃ NCs and about ~ 17 nm size cubic CsPbBr₃ NCs. Therefore, in the lines 65 and 66 of original manuscript, we wrote:

“We also note an emerging controversy over the contrasting nonlinear two-photon behaviour for CsPbBr₃ NCs where both saturable absorption³⁰ and 2PA^{28,29} at 800 nm were reported.”

We realized our oversight here that it is not appropriate to use nonlinear two-photon behaviour for describing both saturable absorption and 2PA, since saturable absorption here refers to one-photon absorption saturation which is a nonlinear one-photon behaviour. To clearly convey the information and to avoid any further misunderstanding, we have revised the description accordingly.

In the original manuscript, lines 65 and 66 **were changed to:**

“We also note an emerging controversy over the contrasting **nonlinear optical behaviour** for CsPbBr₃ NCs where both saturable absorption³⁰ and 2PA^{28,29} at 800 nm were reported (Table S2).” On page 4, paragraph 2, lines 10-11 of the revised manuscript.

In view of the referee’s comments, we have now added a new table to the SI to allow clear comparison of our Z-scan measurements at 800nm on CsPbBr₃ NCs with the reported results in the literature. Both the manuscript and SI have been revised accordingly to reflect this addition, as shown in the above response.

On page 20, of the revised SI, table S2 **was added**:

Table S2. Comparison between the measured 2PA cross-sections of CsPbBr₃ NCs (~ 9nm, cubic) at 800nm through Z-scan measurements to report results excited at the same wavelength and measured with the same technique.

	Xu, Y. et al. J. Am. Chem. Soc. 138 , 3761-3768 (2016)	Wang, Y. et al. Nano Lett. 16 , 448-453 (2016)	Lu, W.-G. et al. Adv. Optical Mater. , DOI:10.1002/adom.201600322 (2016)	Our results
CsPbBr ₃ NCs	(~9 nm, cubic)	(~9 nm, cubic)	(~5 nm spherical and ~ 17 nm cubic)	(~9 nm, cubic)
Interpretation	Two-photon absorption (2PA)	Two-photon absorption (2PA)	Saturable absorption (SA)	Two-photon absorption (2PA)
2PA cross-section at 800 nm (σ_2)	2.7×10^6 GM	1.2×10^5 GM	N.A.	$(2.0 \pm 0.3) \times 10^6$ GM

Comment 2) In my opinion, this method carries to many uncertainties, which can be one of the causes behind the highest values in 5PA.

First, the MEPL strength is a number that depends on a larger number of experimental parameters.

For example:

The intensity of the laser beam is one of the uncertainties.

Are the pulse-width the same for all wavelengths? Is it really Gaussian and for all wavelengths? The authors must verify these points? Do some autocorrelation to verify the pulse-width and so on.

Response 2

We would like to thank the reviewer for his valuable time and constructive comments. For the laser properties, we have not only referred to the manufacturer's data sheets but have also experimentally verified our laser's pulse widths, and its temporal and spatial profiles at various wavelengths used in this work. Detailed discussion is as follow:

The laser system used in our measurements is a femtosecond amplified-pulsed laser system. The excitation laser pulses were generated by an optical parametric amplifier (OPA) (OperA-Solo, Coherent) pumped by a regenerative amplified femtosecond Ti:Sapphire laser system (~50 fs, 800 nm, 1 kHz,; Libra, Coherent). The Coherent Libra regenerative amplifier was seeded by a femtosecond Ti:Sapphire oscillator (~50 fs, 80 MHz, Vitesse, Coherent).

Based on the specification data sheet for our Coherent LibraTM Ti:Sapphire Amplifier System, the original 800nm laser pulses follow the Gaussian distribution and have pulse-width (FWHM) about 50 fs (<https://www.coherent.com/lasers/laser/libra-series>). In addition, based on the specification data

sheet of the Coherent, OperA-Solo optical parametric amplifier (OPA), the output laser pulses (1 kHz, 250-2600 nm) from the OPA with about 50fs, 800nm laser pulses for pumping possess pulse-widths about (1 to 1.2) times of the pump laser pulse duration (*i.e.*, about 50 – 60 fs) and follow the Gaussian distribution (<https://www.coherent.com/lasers/laser/opera-solo-ultrafast-optical-parametric-amplifier>). Hence, the laser pulses applied in our measurements (675nm – 2300nm from OperA-Solo OPA and 800nm from Libra Ti:Sapphire Amplifier) should possess pulse widths in the range of about 50 – 60 fs with maximum variation about 20%. Therefore, the variation in the pulse-widths at different wavelengths will not lead to variation of the order of the measured SPA cross-sections in the lead bromide perovskite NCs.

1. A Single-Shot Autocorrelator (SSA) (High Resolution SSA, Coherent) was used to measure the pulse-width of the 800 nm output laser pulses from Libra Ti:Sapphire Amplifier. The table below summarizes the obtained results, which **was added** to the original SI before line 109 (Table S1). An autocorrelated pulse-width of the original 800 nm laser pulses could be acquired from the product between the calibration value and the time width of the autocorrelation signal on the oscilloscope (FWHM), which was about 70 fs as shown in the below table S1. Since the laser pulse follows the Gaussian distribution, a constant factor 0.707 was multiplied to obtain the pulse-width of the original 800 nm laser pulses, which is about 50fs. The measured pulse-width of the original 800 nm laser pulses is in good agreement with the specification provided on the data sheet of the Coherent, Libra Ti:Sapphire Amplifier System.

In the original SI, before line 109, table S1 **was added**:

Table S1 | SSA measurements to determine the pulse-width of the original 800nm laser pulse from the Libra Ti:Sapphire Amplifier System.

O-scope delta (mS)	1.600
Mic Position 1 (mm)	12.950
Mic Position 2 (mm)	12.880
FWHM of SSA Pulse Trace (mS)	0.240
Calibration Value (pS/mS)	0.292
Autocorrelated PW (pS)	0.070
Gaussian PW (pS)	0.050

Since our SSA is only suitable to measure the pulse-width of laser pulses at 800nm, we have further utilized frequency-resolved optical gating (FROG) (Swamp Optics, UPM-8-50) to characterize the pulse-widths of the laser pulses from both the Libra Ti:Sapphire Amplifier (original 800 nm laser beam) and the OperA-Solo OPA in the wavelength range 700 – 1100nm. The characterization wavelength range is limited by the capability of our FROG set-up. The below figure showing the acquired two-dimensional FROG traces, temporal distributions and spectral distributions of the laser pulses in the wavelength range 700 – 1100nm **was added** to the SI before line 109 and after table S1 (new figure S3 in the revised SI). Figure S3a, d, g, j, m, p, s, v, y displayed the measured two-dimensional FROG traces of the laser pulses, figure S3b, e, h, k, n, q, t, w, z demonstrated the corresponding temporal distributions and figure S3c, f, i, l, o, r, x, aa displayed the corresponding spectral distributions. The laser pulses at 800 nm was from the Libra Ti:Sapphire Amplifier, whereas the laser pulses at other wavelengths were generated by the OperA-Solo OPA.

As shown in figure S3, both the temporal and the spectral distributions of the laser pulses in the wavelength range 700 – 1100nm can be well-fitted by the Gaussian function, indicating laser pulses in the wavelength range indeed follow the Gaussian temporal distribution. The measured pulse-width of the original 800 nm laser pulses is about 50.6 ± 5.1 fs, which agrees well with the result acquired from SSA and the specification of the Coherent (<https://www.coherent.com/lasers/laser/libra-series>), Libra Ti:Sapphire Amplifier System. Moreover, as summarized in Figure S3ad, the measured pulse-widths of the laser pulses in wavelength range 700 – 1100nm range from 50.6 ± 5.1 fs to 56.3 ± 5.6 fs with variation within 15%. The acquired pulse-width result is in excellent agreement with the specification of the Opera-Solo OPA (<https://www.coherent.com/lasers/laser/opera-solo-ultrafast-optical-parametric-amplifier>). Hence, the variation in the pulse-widths at different wavelengths in the range 700 – 1100nm will not lead to variation of the order of the measured 5PA cross-sections in the lead bromide perovskite NCs.

In addition, figure S3ae demonstrated that the measured spectral bandwidths and temporal pulse-widths followed the time-bandwidth relation of Fourier-transform-limited pulses ($\Delta\lambda_{\text{FWHM}} = (A \times K / c) \times ((\lambda_0)^2 / \Delta\tau_{\text{FWHM}})$, C. Rulliere, *Femtosecond Laser Pulses. Principles and Experiments, Springer, (1998)*), a typical characteristic of ultrashort laser pulses generated by mode-locked lasers. c is the speed of light in vacuum, K is a constant depending only on the pulse shape and equals to 0.441 for Gaussian temporal distribution, A is a scaling factor called time-transform-limit factor. From the best fit to the experimental results, A was determined to be about 1.27, as indicated in figure S3ae.

In the original SI, before line 109 and after the above table S1, figure S3 **was added**:

Figure S3 | Pulse-width and spectral bandwidth measurements of the laser beam at wavelength range 700-1100 nm by FROG. (a, d, g, j, m, p, s, v, y) Measured two-dimensional FROG traces of the laser pulses. x axis represents the delay time and y axis corresponds to the wavelength. The right side is the scale bar characterizing the normalized magnitude of the FROG signal; (b, e, h, k, n, q, t, w, z) The corresponding temporal distributions; (c, f, i, l, o, r, x, aa) The corresponding spectral distributions; (ab) Summarized Ch^2 and R^2 of the Gaussian function fitting to laser pulse temporal distribution at all wavelengths. Both the negligible Ch^2 and the very close to unity R^2 demonstrates the good fitting thus indicating the Gaussian temporal distribution of the laser pulses; (ac) Summarized Ch^2 and R^2 of the Gaussian function fitting to spectral distribution at all wavelengths with similar negligible Ch^2 and very close to unity R^2 , demonstrating the good fitting thus the Gaussian spectral distribution of the laser pulses, which is in agreement with (ab); (ad) Summary of the measured pulse-widths of the laser pulses at all wavelengths; (ae) Relation between the measured spectral bandwidths and temporal pulse-widths, together with the fitting with the time-bandwidth relation of Fourier-transform-limited pulses: $\Delta\lambda_{\text{FWHM}} = (A \times K / c) \times ((\lambda_0)^2 / \Delta\tau_{\text{FWHM}})$.

3. Unfortunately, due to limited available wavelength range of our FROG set-up, we could not perform direct measurements on the pulse-widths of utilized laser pulses at wavelengths beyond the range of 700 – 1100nm (*i.e.*, 675 nm and 1200-2300 nm). Based on the above verified pulse-width-bandwidth relation of Fourier-transform-limited pulses ($\Delta\lambda_{\text{FWHM}} = (A \times K / c) \times ((\lambda_0)^2 / \Delta\tau_{\text{FWHM}})$), C. Rulliere, *Femtosecond Laser Pulses. Principles and Experiments*, Springer, (1998), we have measured the spectral profiles of laser pulses at wavelength range 675 – 1700 nm to indirectly derive the pulse-widths at these wavelengths. The overlapping wavelength range of 700 – 1100 nm using this approach serves as a self-consistent cross-check with the results obtained from the FROG approach. The visible monochromator (Acton, Spectra Pro 2750i) coupled with silicon CCD (Princeton Instruments, Pixis 100B) and infrared monochromator (Acton, Spectra Pro

2300i) coupled with liquid-nitrogen-cooled InGaAs infrared detector (Princeton Instruments, 7490-0001) were employed to acquire the optical spectra of the laser beam. The up-limit of the characterized wavelength range is determined by the capability of our infrared monochromator (Acton, Spectra Pro 2300i) coupled with liquid-nitrogen-cooled InGaAs infrared detector (Princeton Instruments, 7490-0001).

The acquired optical spectra of the laser pulses at wavelength range 675-1700 nm were displayed in the figure below, which **was added** to the SI before line 109 and after the above figure S3 (new figure S4 in the revised SI). Figure S4a-q displayed the measured optical spectra at the wavelengths range from 675 – 1700 nm, respectively. The negligible χ^2 and the very close to unity R^2 at all wavelengths summarized in figure S4r demonstrates the good Gaussian fitting, revealing the Gaussian spectral distribution of the laser pulses at these wavelengths. This is highly consistent with the results acquired from FROG measurements and indicates Gaussian temporal distributions at the wavelength range 675 – 1700 nm, beyond the one obtained from FROG results. Due to the limitation of our experimental set-up, we could not directly measure the optical spectra at wavelength range 1800 – 2300 nm. However, since the laser pulses at wavelength range 1700 – 2300 nm belong to the same idler part of the OperA Solo output, laser pulses at wavelength range 1800 – 2300 nm are expected to possess similar Gaussian temporal and spectral distributions as at 1700 nm.

Figure S4s displayed the comparison between the acquired spectral bandwidths at 700-1100 nm by spectrometer and that obtained from the FROG measurements. The acquired spectral bandwidths at different wavelengths were summarized in figure S4t. The spectral bandwidths obtained at wavelength range 700 – 1000 nm are in excellent agreement with the ones measured by FROG, with variation less than 5%. Moreover, the derived pulse-widths at wavelengths of 675 & 1200 – 1700 nm based on the verified pulse-width-bandwidth relation of Fourier-transform-limited pulses and its comparison with the measured pulse-widths at 700 – 1100 nm were shown in figure S4t. The pulse widths at wavelength range 675 – 1700 nm are in the range of $(50.6 \pm 5.1) - (62.9 \pm 6.3)$ fs, with maximum variation about 25%, consistent with the specification of the OperA-Solo OPA. The measured pulse-widths are slightly stretched at longer wavelengths (maximum around 63 fs) compared to the value we had used in our original calculations (50 fs). This will lead to an overestimation of the applied laser peak intensities and consequently an underestimation of the multi-photon absorption cross-sections at longer wavelengths for our reported values. For example, let us consider that the actual pulse widths are in fact longer at 63 fs for five-photon excitations instead of 50 fs used in the manuscript (*i.e.*, a pulse width stretch of about 25%). Based on the expression of the five-photon excited upconversion fluorescence of $(\eta\sigma_5) \propto F_5/(\tau_{FWHM}I_0^5) \propto 1/(\tau_{FWHM} \cdot 1/(\tau_{FWHM})^5) \propto (\tau_{FWHM})^4$ (Xu, C. *et al. J. Opt. Soc. Am. B* **13**, 481-491, (1996); Maiti, S. *et al. Science* **275**, 530-532 (1997) and eq. (7) in the original SI, τ_{FWHM} is the pulse width, I_0 is the laser peak intensity and F_5 is the five-photon excited upconversion PL strength), the five-photon action cross-section ($\eta\sigma_5$) and 5PA cross-sections (σ_5) would be in fact larger than our reported value (*i.e.*, about 2.5 times larger). Therefore, our reported 5PA cross-sections are conservative values.

Measurements of the spectral profile at wavelengths in the range 1800 – 2300 nm are beyond the capability of our experimental set-up. However, as mentioned, laser beams at wavelength range 1700 – 2300 nm belong to the same idler part of the OperA Solo OPA output. Therefore, laser pulses at wavelength range 1800 – 2300 nm are expected to have similar or slightly more stretched

pulse duration, which is also in agreement with the specification of the OperA-Solo OPA that the output laser pulses (1 kHz, 250-2600 nm) from it (with about 50fs, 800nm laser pulses pumping) possess pulse-widths about (1 - 1.2) times of the pump laser pulse duration (*i.e.*, about 50 – 60 fs).

In summary, we have carefully rechecked all the temporal and spectral uncertainties of the applied laser sources highlighted by the reviewer. The applied laser pulses at wavelengths 675 – 2300 nm were demonstrated to possess Gaussian temporal and spectral distribution. And the laser pulses at different wavelengths have similar pulse durations with maximum variation about 25%. The original laser pulse at 800 nm output from the Libra Ti:Sapphire Amplifier possess the minimum pulse duration of (50.6 ± 5.1) fs, and the laser pulses output from OperA-Solo OPA are slightly stretched. In addition, the laser pulses at longer wavelengths were found to have slightly more stretched pulse duration than the value used in our original manuscript. This will lead to overestimation of the applied laser peak intensities and consequently an underestimation of the multi-photon absorption cross-sections at longer wavelengths in our reported values (which are 2.5 times smaller). Nonetheless, such underestimation will not change the order of the measured 5PA cross-sections. Our reported 5PA cross-sections are in fact conservative values.

In the original SI, before line 109 and after the above figure S3, figure S4 **was added**:

Figure S4 | Spectral bandwidth measurements of the laser beam at wavelength range 675-1700 nm by a visible monochromator (Acton, Spectra Pro 2750i) coupled with silicon CCD (Princeton Instruments, Pixis 100B) and infrared monochromator (Acton, Spectra Pro 2300i) coupled with liquid-nitrogen-cooled InGaAs infrared detector (Princeton Instruments, 7490-0001), together with the pulse derivation. (a-q) Measured optical spectra at the wavelengths range from 675 nm to 1700nm; **(r)** Summarized Ch^2 and R^2 of the Gaussian function fitting to laser pulse spectral distribution at wavelengths 675-1700 nm. Both the negligible Ch^2 and the very close to unity R^2 demonstrate the good fitting, thus indicating the Gaussian spectral distribution of the laser pulses; **(s)** Comparison of the acquired spectral bandwidths at 700-1100 nm by spectrometer with that obtained from the FROG measurements, indicating high consistency with variation less than 5%; **(t)** Summary of the measured spectral bandwidths at 675-1700 nm and the comparison of the derived pulse-widths at wavelengths of 675 & 1200 - 1700 nm based on the pulse-width-bandwidth relation of Fourier-transform-limited pulses with the measured pulse-widths at 700-1100 nm by FROG, revealing the pulse widths at wavelength range 675 – 1700 nm are in the range of $(50.6 \pm 5.1) - (62.9 \pm 6.3)$ fs, with variation smaller than 25%, consistent with the specification of the OperA-Solo OPA.

4. We have also performed knife-edge beam scans (Arnaud, J.A. *et al. Appl. Opt.* **10**, 2775-2776 (1971); Khosrofiyan, J.M. *et al. Appl. Opt.* **22**, 3406-3410 (1983)) along both the x - and y -directions of the cross-sectional plane of the laser beam at different wavelengths to characterize its spatial profile. In the measurements, as the knife-edge was moved along both the x - and y -directions of cross-section plane of laser beams, a power/energy meter (for measuring the average power or pulse energy of ultrashort laser pulses - Coherent, LabMax-Top) was applied to measure the total transmitted energy as a function of the scan distance along the x - or y -direction (*i.e.*, x - or y -position).

Below figure showed the acquired knife-edge scan curves along both the x - and y -direction at wavelengths in the range 675-2300 nm, which was added to the SI before the original line 109 and after the above figure S4 (new figure S5 in the new SI). The measured normalized knife-edge scan curves along both the x - and y -direction at all wavelengths was displayed in Figure S5a-y, together with the Gaussian fitting (Arnaud, J.A. *et al. Appl. Opt.* **10**, 2775-2776 (1971); Khosrofiyan, J.M. *et*

al. *Appl. Opt.* **22**, 3406-3410 (1983); de Araújo, M. A. *et al. Appl. Opt.* **48**, 393-396 (2009)). The acquired energy at each point (x or y) is the average of ten measured values with a time interval of 0.5s. The expression for the Gaussian fitting is: $E(x)/E_{max} = E_0/E_{max} + 1/2(1 + erf(\sqrt{2}(x - x_0)/w))$, similar expression also applies to the y -direction by simply exchanging x with y (Arnaud, J.A. *et al. Appl. Opt.* **10**, 2775-2776 (1971); Khosrofiyan, J.M. *et al. Appl. Opt.* **22**, 3406-3410 (1983); de Araújo, M. A. *et al. Appl. Opt.* **48**, 393-396 (2009)). $E(x)$ is measured laser energy at scan distance x , E_{max} is the maximum energy and E_0 is background energy, x_0 is a position of shift with the half of the real energy, erf is a standard error function and w is the measured laser beam radius ($1/e^2$ radius). The acquired knife-edge scan curves in both x - and y -directions at all wavelengths were well-fitted using the above Gaussian expression, as shown in Figure S5a-y. Figure S5z, aa summarized the negligible χ^2 and the very close to unity R^2 for the Gaussian fitting at all wavelengths in both the x - and y -direction, indicating excellent Gaussian fitting and consequently the Gaussian spatial distribution of laser beams along both directions at all wavelengths. Moreover, the high consistency between the measured Knife-edge scans in the x direction and y direction at all wavelengths manifested in Figure S5a-y reveals the two-dimension Gaussian spatial distribution of the laser beams in the cross-section plane. Figure S5ab summarized the acquired laser beam radii along both x and y directions at all wavelengths. The measured laser beam radii in x -direction at all wavelengths are in good agreement with the ones measured by in y -direction with variation less than 6 %, further demonstrating that the two-dimension spatial profiles of the laser beams in the cross-sectional plane follow the Gaussian distribution. Hence, the knife edge beam scans along both the x - and y -directions of cross-sectional plane of laser beams at different wavelengths validate that spatial profiles of the applied laser beams followed the Gaussian distribution.

In addition, as in figure S5ab, the laser beam radius at all wavelengths are in range of $(1.8 \pm 0.2) - (4.1 \pm 0.4)$ mm (*i.e.*, laser beam diameters $(3.6 \pm 0.4) - (8.2 \pm 0.8)$ mm). Laser beam output from the Libra Ti:Sapphire Amplifier at 800 nm was found to have the largest beam radius of (4.1 ± 0.4) mm. Within each different laser output component from the OperA Solo OPA (*i.e.*, second harmonic signal, second harmonic idler, single and idler), the acquired beam radius increases with increasing wavelengths. In addition, for all the laser beams from the OperA Solo OPA, the acquired beam radii followed the overall increasing trend with increasing wavelengths.

In summary, we have carefully rechecked the spatial uncertainties of the applied laser sources highlighted by the reviewer. Through the knife-edge beam scans along both the x - and y -directions of cross-sectional plane of laser beams, the spatial profiles of the applied laser beams at wavelengths in the range 675-2300 nm were validated to follow the Gaussian distribution. The measured laser beam radii at all wavelengths are in range of $(1.8 \pm 0.2) - (4.1 \pm 0.4)$ mm. The maximum beam radius was at 800 nm from the Libra Ti:Sapphire Amplifier and beam radii of the laser beams from OperA Solo OPA follow an overall increasing trend with increasing wavelengths.

In the original SI, before line 109 and after the above figure S4, figure S5 **was added**:

Figure S5 | Knife-edge scans along the x - and y -directions of the cross-sectional plane of the laser beam for characterizing the spatial distributions of the laser beam and measuring its beam radius. (a-y) Measured knife-edge scans curves along the x - and y -directions at the excitation wavelengths range from 675 nm to 2300nm, together with the Gaussian distribution fitting; (z,aa) Summarized χ^2 and R^2 of the Gaussian fitting to laser spatial distributions along both x - and y -directions at wavelengths 675-2300 nm. Both the negligible χ^2 and the very close to unity R^2 demonstrate the good fitting thus indicating the Gaussian spatial distribution of the laser beam at both x - and y -directions ; (ab) Summary of the measured laser beam radii along both x - and y -directions at 675-2300 nm. The high consistency of the measured laser beam radii in x -direction with that measured in y -direction at all wavelengths with variation less than 6 %, further validating that the two-dimension spatial profiles of the laser beams in the cross-sectional plane follow the Gaussian distributions.

In view of the reviewer's comment, in the original SI, after line 108, the related discussion on the temporal, spectral and spatial distributions of the applied femtosecond laser beams at various wavelengths **was added** on page 6 of the revised SI:

“Characterization of the temporal, spectral and spatial distributions of the applied femtosecond laser beams at various wavelengths

(1) Characterization of the temporal, spectral distributions of the applied femtosecond laser beams at various wavelengths

(i) Specifications from the manufacturer’s data sheets

The excitation laser pulses applied in our measurements were generated by an optical parametric amplifier (OPA) (OperA-Solo, Coherent) pumped by a regenerative amplified femtosecond Ti:Sapphire laser system (~50 fs, 800 nm, 1 kHz; Libra, Coherent). The Coherent Libra regenerative amplifier was seeded by a femtosecond Ti:Sapphire oscillator (~50 fs, 80 MHz, Vitesse, Coherent).

Based on the specification data sheet for Coherent, Libra™ Ti:Sapphire Amplifier System, the original 800nm laser pulses follow the Gaussian distribution and have pulse-width (FWHM) about 50 fs¹⁴. In addition, based on the specification data sheet of the Coherent, OperA-Solo optical parametric amplifier (OPA), the output laser pulses (1 kHz, 250-2600 nm) from the OPA with about 50fs, 800nm laser pulses for pumping possess pulse-widths about (1 to 1.2) times of the pump laser pulse duration (*i.e.*, about 50 – 60 fs) and follow the Gaussian distribution¹⁵. Hence, the laser pulses applied in our measurements (675nm – 2300nm from OperA-Solo OPA and 800nm from Libra Ti:Sapphire Amplifier) should possess pulse widths in the range of about 50 – 60 fs with maximum variation about 20%. Therefore, the variation in the pulse-widths at different wavelengths will not lead to variation of the order of the measured 5PA cross-sections in the lead bromide perovskite NCs.

(ii) Temporal distribution and pulse width characterization for laser beam at 800 nm utilizing a Single-Shot Autocorrelator (SSA)

A Single-Shot Autocorrelator (SSA) (High Resolution SSA, Coherent) was used to measure the pulse-width of the 800 nm output laser pulses from Libra Ti:Sapphire Amplifier. Table S1 summarizes the obtained results. An autocorrelated pulse-width of the original 800 nm laser pulses could be acquired from the product between the calibration value and the time width of the autocorrelation signal on the oscilloscope (FWHM), which was about 70 fs as shown in the below table S1. Since the laser pulse follows the Gaussian distribution, a constant factor 0.707 was multiplied to obtain the pulse-width of the original 800 nm laser pulses, which is about 50fs. The measured pulse-width of the original 800 nm laser pulses is in good agreement with the specification provided on the data sheet of the Coherent, Libra Ti:Sapphire Amplifier System.

(iii) Direct characterization of the temporal and spectral distributions of the applied femtosecond laser beams at 700-1100 nm utilizing a frequency-resolved optical gating (FROG)

Since our SSA is only suitable to measure the pulse-width of laser pulses at 800nm, we have further utilized frequency-resolved optical gating (FROG) (Swamp Optics, UPM-8-50) to characterize the pulse-widths of the laser pulses from both the Libra Ti:Sapphire Amplifier (original 800 nm laser beam) and the OperA-Solo OPA in the wavelength range 700 – 1100nm. The characterization wavelength range is limited by the capability of our FROG set-up. Figure S3 shows the acquired two-dimensional FROG traces, temporal distributions and spectral distributions of the laser pulses in the wavelength range 700 – 1100nm. Figure S3a, d, g, j, m, p, s, v, y displayed the measured two-dimensional FROG traces of the laser pulses, figure S3b, e, h, k, n, q, t, w, z demonstrated the corresponding temporal distributions and figure S3c, f, i, l, o, r, x, aa displayed the corresponding

spectral distributions. The laser pulses at 800 nm was from the Libra Ti:Sapphire Amplifier, whereas the laser pulses at other wavelengths were generated by the OperA-Solo OPA.

As shown in figure S3, both the temporal and the spectral distributions of the laser pulses in the wavelength range 700 – 1100nm can be well-fitted by the Gaussian function, indicating laser pulses in the wavelength range indeed follow the Gaussian temporal distribution. The measured pulse-width of the original 800 nm laser pulses is about 50.6 ± 5.1 fs, which agrees well with the result acquired from SSA and the specification of the Coherent, Libra Ti:Sapphire Amplifier System¹⁴. Moreover, as summarized in Figure S3ad, the measured pulse-widths of the laser pulses in wavelength range 700 – 1100nm range from 50.6 ± 5.1 fs to 56.3 ± 5.6 fs with variation within 15%. The acquired pulse-width result is in good agreement with the specification of the OperA-Solo OPA¹⁵. Hence, the variation in the pulse-widths at different wavelengths in the range 700 – 1100nm will not lead to variation of the order of the measured 5PA cross-sections in the lead bromide perovskite NCs.

In addition, figure S3ae demonstrated that the measured spectral bandwidths and temporal pulse-widths followed the time-bandwidth relation of Fourier-transform-limited pulses ($\Delta\lambda_{\text{FWHM}} = (A \times K / c) \times ((\lambda_0)^2 / \Delta\tau_{\text{FWHM}})$)¹⁶, a typical characteristic of ultrashort laser pulses generated by mode-locked lasers. c is the speed of light in vacuum, K is a constant depending only on the pulse shape and equals to 0.441 for Gaussian temporal distribution, A is a scaling factor called time-transform-limit factor. From the best fit to the experimental results, A was determined to be about 1.27, as indicated in figure S3ae.

(iv) Direct characterization of the spectral distributions and indirect characterization of the temporal distributions of the applied femtosecond laser beams at 675-1700 nm utilizing a visible monochromator coupled with a CCD and an infrared monochromator coupled with a liquid-nitrogen-cooled InGaAs infrared detector

Unfortunately, due to limited available wavelength range of our FROG set-up, we could not perform direct measurements on the pulse-widths of utilized laser pulses at wavelengths beyond the range of 700 – 1100nm (*i.e.*, 675 and 1200-2300 nm). Based on the above verified pulse-width-bandwidth relation of Fourier-transform-limited pulses ($\Delta\lambda_{\text{FWHM}} = (A \times K / c) \times ((\lambda_0)^2 / \Delta\tau_{\text{FWHM}})$)¹⁶, we have measured the spectral profiles of laser pulses at wavelength range 675 – 1700 nm to indirectly derive the pulse-widths at these wavelengths. The overlapping wavelength range of 700 – 1100 nm using this approach serves as a self-consistent cross-check with the results obtained from the FROG approach. The visible monochromator (Acton, Spectra Pro 2750i) coupled with CCD (Princeton Instruments, Pixis 100B) and infrared monochromator (Acton, Spectra Pro 2300i) coupled with liquid-nitrogen-cooled InGaAs infrared detector (Princeton Instruments, 7490-0001) were employed to acquire the optical spectra of the laser beam. The upper-limit of the characterized wavelength range is determined by the capability of our infrared monochromator (Acton, Spectra Pro 2300i) coupled with liquid-nitrogen-cooled InGaAs infrared detector (Princeton Instruments, 7490-0001).

The acquired optical spectra of the laser pulses at wavelength range 675-1700 nm were displayed in figure S4. Figure S4a-q displayed the measured optical spectra at the wavelengths range from 675 – 1700 nm, respectively. The negligible χ^2 and the very close to unity R^2 at all wavelengths summarized in figure S4r demonstrates the good Gaussian fitting, revealing the Gaussian spectral distribution of the laser pulses at these wavelengths. This is highly consistent with the results acquired from FROG measurements and indicates Gaussian temporal distributions at the wavelength range 675 – 1700 nm, beyond the one obtained from FROG results. Due to the limitation of our experimental set-up, we could not directly measure the optical spectra at wavelength range 1800 – 2300 nm. However,

since the laser pulses at wavelength range 1700 – 2300 nm belong to the same idler part of the OperA Solo output, laser pulses at wavelength range 1800 – 2300 nm are expected to possess similar Gaussian temporal and spectral distributions as at 1700 nm.

Figure S4s displayed the comparison between the acquired spectral bandwidths at 700-1100 nm by spectrometer and that obtained from the FROG measurements. The acquired spectral bandwidths at different wavelengths were summarized in figure S4t. The spectral bandwidths obtained at wavelength range 700 – 1000 nm are in excellent agreement with the ones measured by FROG, with variation less than 5%. Moreover, the derived pulse-widths at wavelengths of 675 & 1200 - 1700 nm based on the verified pulse-width-bandwidth relation of Fourier-transform-limited pulses and its comparison with the measured pulse-widths at 700 – 1100 nm were shown in figure S4t. The pulse widths at wavelength range 675 – 1700 nm are in the range of $(50.6 \pm 5.1) - (62.9 \pm 6.3)$ fs, with maximum variation about 25%, consistent with the specifications of the OperA-Solo OPA. The measured pulse-widths are slightly stretched at longer wavelengths (maximum around 63 fs) compared to the value we had used in our original calculations (50 fs). This will lead to an overestimation of the applied laser peak intensities and consequently an underestimation of the multi-photon absorption cross-sections at longer wavelengths for our reported values. For example, let us consider that the actual pulse widths are in fact longer at 63 fs for five-photon excitations instead of 50 fs used in the manuscript (*i.e.*, a pulse width stretch of about 25%). Based on the expression of the five-photon excited upconversion fluorescence of $(\eta\sigma_5) \propto F_5/(\tau_{FWHM}I_0^5) \propto 1/(\tau_{FWHM} \cdot 1/(\tau_{FWHM})^5) \propto (\tau_{FWHM})^4$ (Refs. [17,18] and eq. (8), τ_{FWHM} is the pulse width, I_0 is the laser peak intensity and F_5 is the five-photon excited upconversion PL strength), the five-photon action cross-section ($\eta\sigma_5$) and 5PA cross-sections (σ_5) would be in fact larger than our reported value (*i.e.*, about 2.5 times larger). Therefore, our reported 5PA cross-sections are conservative values.

Measurements of the spectral profile at wavelengths in the range 1800 – 2300 nm are beyond the capability of our experimental set-up. However, as mentioned, laser beams at wavelength range 1700 – 2300 nm belong to the same idler part of the OperA Solo OPA output. Therefore, laser pulses at wavelength range 1800 – 2300 nm are expected to have similar or slightly more stretched pulse duration, which is also in agreement with the specification of the OperA-Solo OPA that the output laser pulses (1 kHz, 250-2600 nm) from it (with about 50fs, 800nm laser pulses pumping) possess pulse-widths about (1 - 1.2) times of the pump laser pulse duration (*i.e.*, about 50 – 60 fs).

In summary, we have carefully rechecked all the temporal and spectral uncertainties of the applied laser sources highlighted by the reviewer. The applied laser pulses at wavelengths 675 – 2300 nm were demonstrated to possess Gaussian temporal and spectral distribution. And the laser pulses at different wavelengths have similar pulse durations with maximum variation about 25%. The original laser pulse at 800 nm output from the Libra Ti:Sapphire Amplifier possess the minimum pulse duration of (50.6 ± 5.1) fs, and the laser pulses output from OperA-Solo OPA are slightly stretched. In addition, the laser pulses at longer wavelengths were found to have slightly more stretched pulse duration, which will lead to overestimation of the applied laser peak intensities and consequently an underestimation of the multi-photon absorption cross-sections at longer wavelengths in our reported values. Nonetheless, such underestimation will not change the order of the measured 5PA cross-sections. Our reported 5PA cross-sections are in fact conservative values.

(2) Characterization of the spatial distributions of the applied femtosecond laser beams at various wavelengths using knife-edge scan method

Knife-edge beam scans^{19,20} was performed along both the x - and y -directions of the cross-sectional plane of the laser beam at different wavelengths to characterize its spatial profile. In the measurements, as the knife-edge was moved along both the x - and y -directions of cross-section plane of laser beams, a power/energy meter (for measuring the average power or pulse energy of ultrashort laser pulses - Coherent, LabMax-Top) was applied to measure the total transmitted energy as a function of the scan distance along the x - or y -direction (*i.e.*, x - or y -position).

Figure S5 shows the acquired knife-edge scan curves along both the x - and y -direction at wavelengths in the range 675-2300 nm. The measured normalized knife-edge scan curves along both the x - and y -direction at all wavelengths is displayed in figure S5a-y, together with the Gaussian fitting¹⁹⁻²¹. The acquired energy at each point (x or y) is the average of ten measured values with a time interval of 0.5s. The expression for the Gaussian fitting is: $E(x)/E_{max} = E_0/E_{max} + 1/2(1 + erf(\sqrt{2}(x - x_0)/w))$, similar expression also applies to the y -direction by simply exchanging x with y ¹⁹⁻²¹. $E(x)$ is measured laser energy at scan distance x , E_{max} is the maximum energy and E_0 is background energy, x_0 is a position of shift with the half of the real energy, erf is a standard error function and w is the measured laser beam radius ($1/e^2$ radius). The acquired knife-edge scan curves in both x - and y -directions at all wavelengths were well-fitted using the above Gaussian expression, as shown in figure S5a-y. Figure S5z, aa summarized the negligible χ^2 and the very close to unity R^2 for the Gaussian fitting at all wavelengths in both the x - and y -direction, indicating excellent Gaussian fitting and consequently the Gaussian spatial distribution of laser beams along both directions at all wavelengths. Moreover, the high consistency between the measured knife-edge scans in the x direction and y direction at all wavelengths manifested in figure S5a-y reveals the two-dimension Gaussian spatial distribution of the laser beams in the cross-section plane. Figure S5ab summarized the acquired laser beam radiuses along both x and y directions at all wavelengths. The measured laser beam radii in x -direction at all wavelengths are in good agreement with the ones measured by in y -direction with variation less than 6 %, further demonstrating that the two-dimension spatial profiles of the laser beams in the cross-sectional plane follow the Gaussian distribution. Hence, the knife edge beam scans along both the x - and y -directions of cross-sectional plane of laser beams at different wavelengths validate that spatial profiles of the applied laser beams followed the Gaussian distribution.

In addition, as in figure S5ab, the laser beam radius at all wavelengths are in range of $(1.8 \pm 0.2) - (4.1 \pm 0.4)$ mm (*i.e.*, laser beam diameters $(3.6 \pm 0.4) - (8.2 \pm 0.8)$ mm). Laser beam output from the Libra Ti:Sapphire Amplifier at 800 nm was found to have the largest beam radius of (4.1 ± 0.4) mm. Within each different laser output component from the OperA Solo OPA (*i.e.*, second harmonic signal, second harmonic idler, single and idler), the acquired beam radius increases with increasing wavelengths. In addition, for all the laser beams from the OperA Solo OPA, the acquired beam radii followed the overall increasing trend with increasing wavelengths.

In summary, we have carefully rechecked the spatial uncertainties of the applied laser sources highlighted by the reviewer. Through the knife-edge beam scans along both the x - and y -directions of cross-sectional plane of laser beams, the spatial profiles of the applied laser beams at wavelengths in the range 675-2300 nm were validated to follow the Gaussian distribution. The measured laser beam radii at all wavelengths are in range of $(1.8 \pm 0.2) - (4.1 \pm 0.4)$ mm. The maximum beam radius was at 800 nm from the Libra Ti:Sapphire Amplifier and beam radii of the laser beams from OperA Solo OPA follow an overall increasing trend with increasing wavelengths.”

New references 14-21 **were added** to the SI:

14. <https://www.coherent.com/lasers/laser/libra-series>.
15. <https://www.coherent.com/lasers/laser/opera-solo-ultrafast-optical-parametric-amplifier>.
16. Rulliere, C. Femtosecond Laser Pulses. Principles and Experiments, *Springer*, (1998).
17. Xu, C. & Webb, W. W. Measurement of two-photon excitation cross sections of molecular fluorophores with data from 690 to 1050 nm. *J. Opt. Soc. Am. B* **13**, 481-491, (1996).
18. Maiti, S., Shear, J. B., Williams, R. M., Zipfel, W. R. & Webb, W. W. Measuring Serotonin Distribution in Live Cells with Three-Photon Excitation. *Science* **275**, 530-532 (1997).
19. Arnaud, J. A. *et al.* Technique for Fast Measurement of Gaussian Laser Beam Parameters. *Appl. Opt.* **10**, 2775-2776, (1971).
20. Khosrofian, J. M. & Garetz, B. A. Measurement of a Gaussian laser beam diameter through the direct inversion of knife-edge data. *Appl. Opt.* **22**, 3406-3410, (1983).
21. de Araújo, M. A., Silva, R., de Lima, E., Pereira, D. P. & de Oliveira, P. C. Measurement of Gaussian laser beam radius using the knife-edge technique: improvement on data analysis. *Appl. Opt.* **48**, 393-396, (2009).

Moreover, in view of the reviewer's comments, the related discussion on properties of the excitation laser sources was revised accordingly to add the discussion on the temporal, spectral and spatial distributions of the applied laser source, as follows:

In the Methods section of the original manuscript, lines 301-304: “*The excitation laser pulses (~ 50 fs, 1 kHz, 250-2600 nm) were generated by an optical parametric amplifier (OPA) (OperA-Solo, Coherent) pumped by a regenerative amplified femtosecond Ti:Sapphire laser system (~50 fs, 800 nm, 1 kHz,; Libra, Coherent). The Coherent Libra regenerative amplifier was seeded by a femtosecond Ti:Sapphire oscillator (~50 fs, 80 MHz, Vitesse, Coherent).*”

was revised to:

“The excitation laser pulses (~ 50 fs, 1 kHz, 250-2600 nm) were generated by an optical parametric amplifier (OPA) (OperA-Solo, Coherent) pumped by a regenerative amplified femtosecond Ti:Sapphire laser system (~50 fs, 800 nm, 1 kHz,; Libra, Coherent). The Coherent Libra regenerative amplifier was seeded by a femtosecond Ti:Sapphire oscillator (~50 fs, 80 MHz, Vitesse, Coherent). The temporal, spectral and spatial profiles of the applied excitation laser source at wavelengths 675-2300nm follow Gaussian distribution based on the specifications in the manufacturer's data sheets, and the pulse widths at different wavelengths are in the range of 50-60 fs. Moreover, the Gaussian distribution of the temporal, spectral profiles of the applied excitation laser source at wavelengths 675-2300nm was experimentally verified by the characterizations with single-shot autocorrelator (SSA) (High Resolution SSA, Coherent), frequency-resolved optical gating (FROG) (Swamp Optics, UPM-8-50), visible monochromator (Acton, Spectra Pro 2750i) coupled with CCD (Princeton Instruments, Pixis 100B) and infrared monochromator (Acton, Spectra Pro 2300i) coupled with liquid-nitrogen-cooled InGaAs infrared detector (Princeton Instruments, 7490-0001) (see SI for more details). Additionally, the pulse-widths at different wavelengths were measured to be in the range of $(50.6 \pm 5.1) - (62.9 \pm 6.3)$ fs, highly consistent with the specifications in the data sheet. Moreover, through applying knife-edge scans along both the *x*- and *y*-directions of the cross-sectional planes of the laser beams, the two-dimensional spatial profiles of the laser beams at wavelengths 675-2300nm were validated to follow Gaussian distribution (see SI for more details). ”on page 21, paragraph 2 line 5 of the revised manuscript

To exemplify the superior five-photon excited upconversion PL of the perovskite NCs, we have also performed a direct comparison between the five-photon excited PL from MAPbBr₃/(OA)₂PbBr₄ NCs (~2.1 μM) and from R6G (~ 2.1 μM) under the same experimental conditions. R6G has been well demonstrated to be a nonlinear optical active organic dye with relatively large multi-photon absorption cross-sections and high PLQY (Albota, M. A. *et al. Appl. Opt.* **37**, 7352-7356, (1998); Makarov, N. S. *et al. Opt. Express* **16**, 4029-4047, (2008)). Since there have been no report on the five-photon absorption in R6G, we only provide a qualitatively direct comparison on the five-photon excited PL photos. The unedited photos (taken with the same camera and the same exposure) below show the five-photon excited PL from MAPbBr₃/(OA)₂PbBr₄ NCs in toluene as compared to R6G in methanol at 2100 nm femtosecond laser excitation. As can be seen from the photos below, under the applied excitation conditions, no five-photon excited PL was observed from the R6G in methanol and no upconversion PL signal could be detected with our visible monochromator (Acton, Spectra Pro 2750i) coupled with CCD. In contrast, relatively bright five-photon excited PL was excited under the same experimental conditions at 2100 nm, validating the much superior five-photon absorption properties of our MAPbBr₃/(OA)₂PbBr₄ NCs than R6G. Moreover, R6G exhibits one-photon absorption peak at around 525nm and 2100 nm is the boundary wavelength for R6G where an admix of contributions from both the 4PA and 5PA processes is possible. This further confirms the outstanding 5PA properties of the MAPbBr₃/(OA)₂PbBr₄ NCs.

In the original SI, after figure S7, figure S13 **was added**:

Samples	Excitation fluence		
	3.63mJ/cm ² Excited at 2100 nm	4.62mJ/cm ² Excited at 2100 nm	5.66mJ/cm ² Excited at 2100 nm
R6G in methanol (~ 2.1 μM)			MAPbBr ₃ /(OA) ₂ PbBr ₄ NCs in toluene (~2.1 μM)			
Figure S13 | Comparison of the photographs of five-photon excited upconversion luminescence from MAPbBr₃/(OA)₂PbBr₄ NCs in toluene with that from R6G in methanol at 2100 nm femtosecond laser excitation and under the same experimental conditions.

In view of the reviewer’s concern, in the original SI, after line 213, the related discussion on the qualitatively direct comparison of the five-photon excited upconversion PL from MAPbBr₃/(OA)₂PbBr₄ NCs in toluene with that from R6G **was added**:

“To exemplify the superior five-photon excited upconversion PL of the perovskite NCs, we have also performed a direct comparison between the five-photon excited PL from MAPbBr₃/(OA)₂PbBr₄ NCs (~2.1 μM) and from R6G (~ 2.1 μM) under the same experimental conditions. R6G has been

well demonstrated to be a nonlinear optical active organic dye with relatively large multi-photon absorption cross-sections and high PLQY^{38,39}. Since there have been no report on the five-photon absorption in R6G, we only provide a qualitatively direct comparison on the five-photon excited PL photos. The unedited photos (taken with the same camera and the same exposure) displayed in figure S13 show the five-photon excited PL from MAPbBr₃/(OA)₂PbBr₄ NCs in toluene as compared to R6G in methanol at 2100 nm femtosecond laser excitation. As in figure S13, under the applied excitation conditions, no five-photon excited PL was observed from the R6G in methanol and no upconversion PL signal could be detected with our visible monochromator (Acton, Spectra Pro 2750i) coupled with CCD. In contrast, relatively bright five-photon excited PL was excited under the same experimental conditions at 2100 nm, validating the much superior five-photon absorption properties of our MAPbBr₃/(OA)₂PbBr₄ NCs than R6G. Moreover, R6G exhibits one-photon absorption peak at around 525nm and 2100 nm is the boundary wavelength for R6G where an admixture of contributions from both the 4PA and 5PA processes is possible. This further confirms the outstanding 5PA properties of the MAPbBr₃/(OA)₂PbBr₄ NCs.”

New references 38,39 **were added** into the SI:

38. Albota, M. A., Xu, C. & Webb, W. W. Two-photon fluorescence excitation cross sections of biomolecular probes from 690 to 960 nm. *Appl. Opt.* **37**, 7352-7356, (1998).
39. Makarov, N. S., Drobizhev, M. & Rebane, A. Two-photon absorption standards in the 550-1600 nm excitation wavelength range. *Opt. Express* **16**, 4029-4047, (2008).

In addition, we **have accordingly revised** the related discussion on the photographs showing frequency-upconverted PL from perovskite NCs to add the discussion on the direct comparison of the five-photon excited upconversion PL from MAPbBr₃/(OA)₂PbBr₄ NCs with that from R6G.

In the original manuscript, lines 120-121: “*Photographs in figure S7 clearly demonstrate the frequency-upconverted PL from NCs when they were irradiated with infrared femtosecond laser pulses.*”

was revised to:

“Photographs in figure S12 clearly demonstrate the frequency-upconverted PL from NCs when they were irradiated with infrared femtosecond laser pulses. Furthermore, the direct comparison between the five-photon excited upconversion PL from MAPbBr₃/(OA)₂PbBr₄ NCs and that from R6G having the same concentration and under the same experimental conditions exemplify the superior 5PA properties of the MAPbBr₃/(OA)₂PbBr₄ NCs (see SI for more details).” on page 7, paragraph 1, line 15 of the revised manuscript.

Comment 3) Another point is the distance of the sample to the focal point, 3.5 cm. At this point, the beam waist will be different for each individual wavelength, which gives a factor on the intensity determination. Thus the authors checked this parameter? It is a crucial point.

Response 3)

We would like to thank the reviewer for his valuable time and constructive comments. Yes, we have in fact taken into consideration this point as highlighted by the reviewer in our initial calculations:

The laser intensity distribution after the focal lens was calculated through the propagation principle of Gaussian beams (Saleh, B. E. A. *et al.* Fundamentals of Photonics. Vol. 2, Wiley: Hoboken, New Jersey, (2007)). The beam waist at the sample point was calculated by: $w(d) = w_0\sqrt{1 + d^2/z_0^2}$. d is the distance of the sample to the focal point. w_0 is the beam waist of laser beam at the focal point. z_0 is the Rayleigh length and is related w_0 by $z_0 = \pi w_0^2/\lambda_0$. w_0 is related to the incident laser beam waist before focal lens w'_0 by $w_0 \approx \lambda_0 f / (\pi n w'_0)$, which is related to the excitation laser wavelength λ_0 . f is the focal length, n is the refractive index of the air $n \approx 1$. As in the original SI, the integration of Δf_n was performed over the laser beam intensity distribution with beam waist of $w(d)$ at the sample point as follows:

$$F_n = \iiint \Delta f_n = \iiint (1/n) \phi \rho \eta \sigma_n [I_i(r, z, t)]^n / (\hbar \omega)^n ds dz dt \quad (3)$$

This could be clearly seen from the derived expressions of the the MEPL signal F_n ($n = 2, 3, 4,$ and 5) for 2PA, 3PA, 4PA and 5PA processes, where the distance d dependence was clearly shown. And $w_0 \approx \lambda_0 f / (\pi n w'_0)$ in the below expressions is related to the excitation wavelength λ_0 .

$$F_2 \approx \pi^{3/2} \phi \rho \eta \sigma_2 L \tau_p \omega_0^2 d^2 I_0^2 / [8\sqrt{2} z_0^2 (\hbar \omega)^2] \quad (4)$$

$$F_3 \approx \pi^{3/2} \phi \rho \eta \sigma_3 L \tau_p \omega_0^2 d^2 I_0^3 / [18\sqrt{3} z_0^2 (\hbar \omega)^3] \quad (5)$$

$$F_4 \approx \pi^{3/2} \phi \rho \eta \sigma_4 L \tau_p \omega_0^2 d^2 I_0^4 / [64 z_0^2 (\hbar \omega)^4] \quad (6)$$

$$F_5 \approx \pi^{3/2} \phi \rho \eta \sigma_5 L \tau_p \omega_0^2 d^2 I_0^5 / [50\sqrt{5} z_0^2 (\hbar \omega)^5] \quad (7)$$

In view of the reviewer's concern, we have revised the related discussion in the original SI:

In the original SI, lines 265-269:

“Taking account of the fact that NCs contained in 2-mm-thick quartz cuvettes were placed 3.5 cm away from focal point of the lens and considering the spatial and temporal profiles of the laser pulses are Gaussian functions, the MEPL signal F_n ($n = 2, 3, 4,$ and 5) for 2PA, 3PA, 4PA and 5PA processes can be derived as:

$$F_2 \approx \pi^{3/2} \phi \rho \eta \sigma_2 L \tau_p \omega_0^2 d^2 I_0^2 / [8\sqrt{2} z_0^2 (\hbar \omega)^2] \quad (4)$$

$$F_3 \approx \pi^{3/2} \phi \rho \eta \sigma_3 L \tau_p \omega_0^2 d^2 I_0^3 / [18\sqrt{3} z_0^2 (\hbar \omega)^3] \quad (5)$$

$$F_4 \approx \pi^{3/2} \phi \rho \eta \sigma_4 L \tau_p \omega_0^2 d^2 I_0^4 / [64 z_0^2 (\hbar \omega)^4] \quad (6)$$

$$F_5 \approx \pi^{3/2} \phi \rho \eta \sigma_5 L \tau_p \omega_0^2 d^2 I_0^5 / [50\sqrt{5} z_0^2 (\hbar \omega)^5] \quad (7)$$

, respectively.” **was changed to**

“The beam waist at the sample point is $w(d) = w_0\sqrt{1 + d^2/z_0^2}$. d is the distance of the sample to the focal point. w_0 is the beam waist of laser beam at the focal point. z_0 is the Rayleigh length and is related to w_0 by $z_0 = \pi w_0^2/\lambda_0$. w_0 is related to the incident laser beam waist before focal lens w'_0 by⁴³

$w_0 \approx \lambda_0 f / (n\pi w'_0)$, which is related to the laser wavelength λ_0 . f is the focal length, n is the refractive index of the air $n \approx 1$. Taking account of the fact that NCs contained in 2-mm-thick quartz cuvettes were placed 3.5 cm away from focal point of the lens (beam waists at the sample point are dependent on the excitation wavelengths) and considering the spatial and temporal profiles of the laser pulses are Gaussian functions, the MEPL signal F_n ($n = 2, 3, 4$, and 5) for 2PA, 3PA, 4PA and 5PA processes can be derived as:

$$F_2 \approx \pi^{3/2} \phi \rho \eta \sigma_2 L \tau_p \omega_0^2 d^2 I_0^2 / [8\sqrt{2} z_0^2 (\hbar\omega)^2] \quad (5)$$

$$F_3 \approx \pi^{3/2} \phi \rho \eta \sigma_3 L \tau_p \omega_0^2 d^2 I_0^3 / [18\sqrt{3} z_0^2 (\hbar\omega)^3] \quad (6)$$

$$F_4 \approx \pi^{3/2} \phi \rho \eta \sigma_4 L \tau_p \omega_0^2 d^2 I_0^4 / [64 z_0^2 (\hbar\omega)^4] \quad (7)$$

$$F_5 \approx \pi^{3/2} \phi \rho \eta \sigma_5 L \tau_p \omega_0^2 d^2 I_0^5 / [50\sqrt{5} z_0^2 (\hbar\omega)^5] \quad (8)$$

, respectively.”

New reference 43 **was added**:

43. Saleh, B. E. A. & Teich, M.C. *et al.* Fundamentals of Photonics. Vol. 2, Wiley: Hoboken, New Jersey, (2007)

Comment 4) Why the author did not performed Z-Scan for the other spectral region, mainly for 2100 nm? Or at list on the 3PA and 4PA region. These regions are at list easily to measure multi-photon absorption with Z-Scan technique, as presented in some references cited on this manuscript. It will help to compare the MEPL method and turns the results stronger by two different experimental procedures in a widely spectral region.

Response 4)

We would like to thank the reviewer for his constructive comments. Multi-photon excited upconversion PL measurements have been demonstrated to provide much higher detection sensitivity for characterizing MPA properties of both inorganic semiconductor NCs and organic molecules than open-aperture Z-scan, especially for higher-order MPA and highly luminescent MPA-active materials (Xu, C. *et al.* *J. Opt. Soc. Am. B* **13**, 481-491 (1996); Xu, C. *et al.* *Proc. Natl. Acad. Sci. U.S.A.* **93**, 10763–10768 (1996); He, G. S. *et al.* *Chem. Rev.* **108**, 1245-1330 (2008)). Due to the low detection efficiency, extremely high excitation peak intensities are required in open-aperture Z-scan for characterizing the high-order multi-photon absorption (*i.e.*, four- and five-photon absorption), which may cause sample damage or introduce other effects (or artifacts) with the extremely high excitation intensities. Consequently, most of the characterizations for the high-order multiphoton absorption (*i.e.*, four- and five-photon absorption) were performed with multi-photon excited upconversion PL measurements (He, G. S. *et al.* *J. Opt. Soc. Am. B* **22**, 2219-2228 (2005); He, G. S. *et al.* *Chem. Rev.* **108**, 1245-1330 (2008); Fan, H. H., *et al.* *J. Am. Chem. Soc.* **134**, 7297-7300 (2012)) and open-aperture Z-scan has been seldom used in this region. In view of the reviewer’s comments, we have now performed the open-aperture Z-scan measurements for the 3PA in our lead bromide perovskite NCs at 1050 and 1100nm. The selection of the characterization wavelengths is based on the limited detection wavelength range of our detection (RkP 465, Laser Probe). Detailed discussions are as follows:

Open-aperture Z-scan measurements for quantifying 3PA cross-sections (σ_3) at 1050 and 1100 nm

The applied experimental set-up for the open-aperture Z-scan measurements on 3PA in the lead bromide perovskite NCs at 1050 and 1100 nm is similar to the one used for 2PA at 800 nm, as detailed in lines 317-329 of the original manuscript, lines 109-125 of the original SI and illustrated in figure S3 in original SI. The calibration of our open-aperture Z-scan measurements for 3PA at 1050 and 1100nm was conducted utilizing a wide-gap semiconductor CdS (0.5-mm thick CdS wafer, single crystal). 3PA of CdS single crystal have been well studied both experimentally (Woodall, M.A. *North Texas State University Dissertation*, Ch.4 pg.138 (1985)) and theoretically (Brandi H.S. et al. *J. Phys. C: Solid State Phys.* **16**, 5929-5936 (1983)) in this wavelength range. Open-aperture Z-scan curves on the CdS wafer under excitation peak intensities of about 23 and 36 GW/cm² at 1050 and 1100nm were displayed in figures S6a and S7a. Based on the open-aperture Z-scan theory for 3PA (He, J. et al. *Opt. Express* **13**, 9235-9247 (2005)), $\ln(1-T_{OA})$ (where T_{OA} is the normalized transmittance along the Z-axis in the open-aperture Z-scan) was plotted as a function of $\ln(I_0)$ (I_0 is the maximum excitation intensity on the Z-axis and $I_0 = I_{00}/(1 + z^2/z_0^2)$) at both wavelengths to demonstrate the presence of 3PA, as shown in the insets of figures S6a and S7a. The linear fit with slopes of about 1.95 and 2.03 are indicative of the occurrence of 3PA processes at both wavelengths, which is as expected since the photon energies of the applied laser beams are between one third and half of the bandgap energy of CdS. The extracted 3PA coefficients $\chi(1050 \text{ nm}) \sim (1.4 \pm 0.2) \times 10^{-2} \text{ cm}^3/\text{GW}^2$ and $\chi(1100 \text{ nm}) \sim (1.3 \pm 0.2) \times 10^{-2} \text{ cm}^3/\text{GW}^2$ agree well with both the experimental report (Woodall, M.A. *North Texas State University Dissertation*, Ch.4 pg.138 (1985)) and theoretical calculation (Brandi H.S. et al. *J. Phys. C: Solid State Phys.* **16**, 5929-5936 (1983)), indicating that our Z-scan set-up is properly calibrated.

Figure S6b,c shows the open-aperture Z-scan responses of 3PA in the toluene solutions of MAPbBr₃ (~2.0 μM) and CsPbBr₃ NCs (~1.0 μM) in 1-mm-thick cuvette under excitation at 1050 nm with peak intensities of ~35.0 and ~50.0 GW/cm². On the other hand, open-aperture Z-scans of 3PA in the MAPbBr₃/(OA)₂PbBr₄ NCs (~2.1 μM) excited at 1050 nm with peak intensities of ~25.0 and ~35.0 GW/cm² are shown in figure S6d. The insets of figure S6b-c shows the corresponding plots of $\ln(1-T_{OA})$ vs. $\ln(I_0)$ together with the linear fits, manifesting the presence of 3PA. The acquired slopes are about 1.82, 1.85 and 1.83, indicating relatively larger deviation from 2 (i.e., 15-20%). Such deviation at 1050 nm is in accordance with the obtained slopes from the excitation fluence dependence of multi-photon excited upconversion PL, as shown in figure 2a of the original manuscript. And the deviation may result from the NCs size inhomogeneity leading to an admixture of contributions from both the 2PA and 3PA processes at this boundary wavelength of 1050nm (lines 138-142 of original manuscript). However, since the deviation is less than 20%, 3PA process still dominates at this boundary wavelength.

For excitation at 1100nm, open-aperture Z-scan curves of 3PA in the MAPbBr₃ and CsPbBr₃ NCs in toluene solution under excitation peak intensities of ~75.0 and ~ 100.0 GW/cm² were also acquired and shown in figure S7b,c. On the other hand, figure S7d shows the open-aperture Z-scan curves of 3PA in MAPbBr₃/(OA)₂PbBr₄ NCs in toluene solution with excitation wavelength at 1100nm and excitation peak intensities of ~ 50.0 and ~ 75.0 GW/cm². Corresponding plots of $\ln(1-T_{OA})$ vs. $\ln(I_0)$ and the linear fits are also added to the insets of figure S7b-d, respectively, to validate the occurrence of the 3PA processes. The slopes obtained at 1100nm are about 1.97, 2.01 and 2.03 for MAPbBr₃, CsPbBr₃ and MAPbBr₃/(OA)₂PbBr₄ NCs, respectively, clearly indicating the only 3PA process at 1100 nm excitation.

Open-aperture Z-scan measurements on the pure toluene solvent were also conducted under the maximum excitation peak intensities (*i.e.*, ~ 50.0 GW/cm² excitation at 1050 nm and ~ 100.0 GW/cm² excitation at 1100 nm) as the control experiment, which were shown in figures S6d and S7d. Almost flat open-aperture Z-scan curves of the toluene at both wavelengths suggest its negligible nonlinear absorption response at the applied excitation conditions.

To acquire the 3PA cross-sections of the NC solutions, the below well-established Z-scan theory (Sutherland, R. L. *et al.* *New York, NY: Marcel Dekker*, Second Edition, Revised and Expanded, (2003)) were applied to fit the normalized open-aperture Z-scan transmittance (considering the Gaussian spatial and temporal distributions of utilized laser beam):

$$T(z) = \frac{1}{\sqrt{\pi} p_0(z)} \int_{-\infty}^{\infty} \ln \left\{ \left[1 + p_0^2(z) e^{-2x^2} \right]^{1/2} + p_0(z) e^{-x^2} \right\} dx$$

$p_0(z) = (2\gamma(1-R)^2 L'_{eff} I_{00}^2 / (1 + z^2/z_0^2)^2)^{1/2}$, $\gamma = N\sigma_3 / (\hbar\omega)^2$. R is reflection from the sample front surface; I_{00} is peak intensity at the focal point of the incident laser beam; $L'_{eff} = L$ at 1050 and 1100 nm and L is sample thickness due to the negligible one-photon absorption at these wavelengths; N is concentration of NCs; $\hbar\omega$ is the incident photon energy and σ_3 is the 3PA cross-section. 3PA cross-sections (σ_3) of the MAPbBr₃, MAPbBr₃/(OA)₂PbBr₄ and CsPbBr₃ NCs at 1050 nm were estimated to be $\sim (2.3 \pm 0.3) \times 10^{-74}$ cm⁶s²/photon², $\sim (20 \pm 3) \times 10^{-74}$ cm⁶s²/photon² and $\sim (6.7 \pm 1.0) \times 10^{-74}$ cm⁶s²/photon², respectively. Moreover, for excitation at 1100 nm, the 3PA cross-sections (σ_3) of the MAPbBr₃, MAPbBr₃/(OA)₂PbBr₄ and CsPbBr₃ NCs were calculated to be $\sim (0.55 \pm 0.08) \times 10^{-74}$ cm⁶s²/photon², $\sim (3.3 \pm 0.5) \times 10^{-74}$ cm⁶s²/photon² and $\sim (1.9 \pm 0.3) \times 10^{-74}$ cm⁶s²/photon².

The obtained σ_3 of perovskite NCs through the open-aperture Z-scan measurements are highly consistent with the results acquired from multi-photon excited upconversion PL measurements (figure 2c and table 1 of the original manuscript) with the variation in the range of about 6%-17%. The comparison between the measured σ_3 values from open-aperture Z-scan measurements and from multi-photon excited upconversion PL measurements are summarized in the below table. Hence, the open-aperture Z-scan measurements on 3PA in the perovskite NCs performed at 1050 and 1100 nm further confirm our 3PA results measured with the multi-photon excited upconversion PL technique. The good agreement of the 2PA (as in original SI lines 109-161 and figure S4) and 3PA cross-sections acquired by the multi-photon excited upconversion PL technique with that obtained from open-aperture measurements validates that the multi-photon absorption cross-sections of our perovskite NCs have been properly measured with the multi-photon excited upconversion PL technique.

Table, comparison between the acquired 3PA cross-sections of perovskite NCs by the multi-photon excited upconversion PL technique with that obtained from open-aperture measurements

Perovskite NCs Techniques	3PA $\eta\sigma_3$ ($10^{-74}\text{cm}^6\text{s}^2/\text{photon}^2$) $\lambda_{\text{ex}} = 1050\text{nm}$			3PA $\eta\sigma_3$ ($10^{-74}\text{cm}^6\text{s}^2/\text{photon}^2$) $\lambda_{\text{ex}} = 1100\text{nm}$		
	MAPbBr ₃	MAPbBr ₃ /(OA) ₂ PbBr ₄	CsPbBr ₃	MAPbBr ₃	MAPbBr ₃ /(OA) ₂ PbBr ₄	CsPbBr ₃
	3					3
Open-aperture Z-scan measurements	2.3 ± 0.3	20 ± 3	6.7 ± 1.0	0.55 ± 0.08	3.3 ± 0.5	1.9 ± 0.3
Multi-photon excited upconversion PL measurements	2.7 ± 0.4	22 ± 3	7.6 ± 1.1	0.52 ± 0.08	3.6 ± 0.5	1.7 ± 0.3

In the original SI, after line 169, figures S8 and S9 were added:

Figure S8 | Open-aperture Z-scan measurements on 3PA in perovskite NCs at 1050 nm. (a) Open-aperture Z-scan curves of the standard sample CdS (0.5-mm thick) at 1050 nm with excitation peak intensities of ~ 23 and ~ 36 GW/cm^2 ; (b-c) Open-aperture Z-scan responses from the toluene solutions of MAPbBr₃ (~ 2.0 μM) and CsPbBr₃ NCs (~ 1.0 μM) contained in 1-mm-thick cuvette under laser excitation at 1050 nm and with peak intensities of ~ 35.0 and ~ 50.0 GW/cm^2 ; (d) Open-aperture Z-scan responses from the toluene solutions of MAPbBr₃ MAPbBr₃/(OA)₂PbBr₄ (~ 2.1 μM) contained in 1-mm-thick cuvette excited at 1050 nm with peak intensities of ~ 25.0 and ~ 35.0 GW/cm^2 . The almost flat open-aperture Z-scan curve of the toluene under the

same excitation condition (contained in 1-mm-thick cuvette, laser excitation at 1050 nm with peak intensity of $\sim 50.0 \text{ GW/cm}^2$) is also shown in (d).

Figure S9 | Open-aperture Z-scan measurements on 3PA in perovskite NCs at 1100 nm. (a) Open-aperture Z-scan curves of the standard sample CdS (0.5-mm thick) at 1100 nm with excitation peak intensities of ~ 23 and $\sim 36 \text{ GW/cm}^2$; (b-c) Open-aperture Z-scan responses from the toluene solutions of MAPbBr₃ ($\sim 2.0 \mu\text{M}$) and CsPbBr₃ NCs ($\sim 1.0 \mu\text{M}$) contained in 1-mm-thick cuvette under laser excitation at 1100 nm and with peak intensities of ~ 75.0 and $\sim 100.0 \text{ GW/cm}^2$; (d) Open-aperture Z-scan responses from the toluene solutions of MAPbBr₃ MAPbBr₃/(OA)₂PbBr₄ ($\sim 2.1 \mu\text{M}$) contained in 1-mm-thick cuvette excited at 1100 nm with peak intensities of ~ 50.0 and $\sim 75.0 \text{ GW/cm}^2$. The almost flat open-aperture Z-scan curve of the toluene under the same excitation condition (contained in 1-mm-thick cuvette, laser excitation at 1100 nm with peak intensity of $\sim 100.0 \text{ GW/cm}^2$) is also shown in (d).

In view of the reviewer's concern, we **have added** the related discussion on the open-aperture Z-scan measurements on the 3PA in the perovskite NCs to the original SI, after line 168:

“Open-aperture Z-scan measurements for quantifying 3PA cross-sections (σ_3) at 1050 and 1100 nm

The applied experimental set-up for the open-aperture Z-scan measurements on 3PA in the lead bromide perovskite NCs at 1050 and 1100 nm is similar to the one used for 2PA at 800 nm. The calibration of our open-aperture Z-scan measurements for 3PA at 1050 and 1100nm was conducted utilizing a wide-gap semiconductor CdS (0.5-mm thick CdS wafer, single crystal). 3PA of CdS single crystal have been well studied both experimentally³⁰ and theoretically³¹ in this wavelength range. Open-aperture Z-scan curves on the CdS wafer under excitation peak intensities of about 23 and 36 GW/cm^2 at 1050 and 1100nm are displayed in figures S8a and S9a. Based on the open-

aperture Z-scan theory for 3PA³², $\text{Ln}(1-T_{\text{OA}})$ (where T_{OA} is the normalized transmittance along the Z-axis in the open-aperture Z-scan) was plotted as a function of $\text{Ln}(I_0)$ (I_0 is the maximum excitation intensity on the Z-axis and $I_0 = I_{00}/(1 + z^2/z_0^2)$) at both wavelengths to demonstrate the presence of 3PA, as shown in the insets of figures S8a and S9a. The linear fit with slopes of about 1.95 and 2.03 are indicative of the occurrence of 3PA processes at both wavelengths, which is as expected since the photon energies of the applied laser beams are between one third and half of the bandgap energy of CdS. The extracted 3PA coefficients $\gamma(1050 \text{ nm}) \sim (1.4 \pm 0.2) \times 10^{-2} \text{ cm}^3/\text{GW}^2$ and $\gamma(1100 \text{ nm}) \sim (1.3 \pm 0.2) \times 10^{-2} \text{ cm}^3/\text{GW}^2$ agree well with both the experimental report³⁰ and theoretical calculation³¹, indicating that our Z-scan set-up is properly calibrated.

Figure S8b,c shows the open-aperture Z-scan responses of 3PA in the toluene solutions of MAPbBr₃ (~2.0 μM) and CsPbBr₃ NCs (~1.0 μM) in 1-mm-thick cuvette under excitation at 1050 nm with peak intensities of ~35.0 and ~50.0 GW/cm². On the other hand, open-aperture Z-scans of 3PA in the MAPbBr₃/(OA)₂PbBr₄ NCs (~2.1 μM) excited at 1050 nm with peak intensities of ~25.0 and ~35.0 GW/cm² are shown in figure S8d. The insets of figure S8b-c shows the corresponding plots of $\text{Ln}(1-T_{\text{OA}})$ vs. $\text{Ln}(I_0)$ together with the linear fits, manifesting the presence of 3PA. The acquired slopes are about 1.82, 1.85 and 1.83, indicating relatively larger deviation from 2 (*i.e.*, 15-20%). Such deviation at 1050 nm is in accordance with the obtained slopes from the excitation fluence dependence of multi-photon excited upconversion PL, as shown in figure 2a of the original manuscript. And the deviation may result from the NCs size inhomogeneity leading to an admixture of contributions from both the 2PA and 3PA processes at this boundary wavelength of 1050nm (lines 138-142 of original manuscript). However, since the deviation is less than 20%, 3PA process still dominates at this boundary wavelength.

For excitation at 1100nm, open-aperture Z-scan curves of 3PA in the MAPbBr₃ and CsPbBr₃ NCs in toluene solution under excitation peak intensities of ~75.0 and ~ 100.0 GW/cm² were also acquired and shown in figure S9b,c. On the other hand, figure S9d shows the open-aperture Z-scan curves of 3PA in MAPbBr₃/(OA)₂PbBr₄ NCs in toluene solution with excitation wavelength at 1100nm and excitation peak intensities of ~ 50.0 and ~ 75.0 GW/cm². Corresponding plots of $\text{Ln}(1-T_{\text{OA}})$ vs. $\text{Ln}(I_0)$ and the linear fits are also added to the insets of figure S9b-d, respectively, to validate the occurrence of the 3PA processes. The slopes obtained at 1100nm are about 1.97, 2.01 and 2.03 for MAPbBr₃, CsPbBr₃ and MAPbBr₃/(OA)₂PbBr₄ NCs, respectively, clearly indicating the only 3PA process at 1100 nm excitation.

Open-aperture Z-scan measurements on the pure toluene solvent were also conducted under the maximum excitation peak intensities (*i.e.*, ~ 50.0 GW/cm² excitation at 1050 nm and ~ 100.0 GW/cm² excitation at 1100 nm) as the control experiment, which were shown in figures S8d and S9d. Almost flat open-aperture Z-scan curves of the toluene at both wavelengths suggest its negligible nonlinear absorption response at the applied excitation conditions.

To acquire the 3PA cross-sections of the NC solutions, the below well-established Z-scan theory³³ were applied to fit the normalized open-aperture Z-scan transmittance (considering the Gaussian spatial and temporal distributions of utilized laser beam):

$$T(z) = \frac{1}{\sqrt{\pi} p_0(z)} \int_{-\infty}^{\infty} \ln \left\{ \left[1 + p_0^2(z) e^{-2x^2} \right]^{1/2} + p_0(z) e^{-x^2} \right\} dx \quad (3)$$

$p_0(z) = (2\gamma(1-R)^2 L'_{eff} I_{00}^2 / (1+z^2/z_0^2)^2)^{1/2}$, $\gamma = N\sigma_3 / (\hbar\omega)^2$. R is reflection from the sample front surface; I_{00} is peak intensity at the focal point of the incident laser beam; $L'_{eff} = L$ at 1050 and 1100 nm and L is sample thickness due to the negligible one-photon absorption at these wavelengths; N is concentration of NCs; $\hbar\omega$ is the incident photon energy and σ_3 is the 3PA cross-section. 3PA cross-sections (σ_3) of the MAPbBr₃, MAPbBr₃/(OA)₂PbBr₄ and CsPbBr₃ NCs at 1050 nm were estimated to be $\sim (2.3 \pm 0.3) \times 10^{-74}$ cm⁶s²/photon², $\sim (20 \pm 3) \times 10^{-74}$ cm⁶s²/photon² and $\sim (6.7 \pm 1.0) \times 10^{-74}$ cm⁶s²/photon², respectively. Moreover, for excitation at 1100 nm, the 3PA cross-sections (σ_3) of the MAPbBr₃, MAPbBr₃/(OA)₂PbBr₄ and CsPbBr₃ NCs were calculated to be $\sim (0.55 \pm 0.08) \times 10^{-74}$ cm⁶s²/photon², $\sim (3.3 \pm 0.5) \times 10^{-74}$ cm⁶s²/photon² and $\sim (1.9 \pm 0.3) \times 10^{-74}$ cm⁶s²/photon².

The obtained σ_3 of perovskite NCs through the open-aperture Z-scan measurements are highly consistent with the results acquired from multi-photon excited upconversion PL measurements (figure 2c and table 1 of the original manuscript) with the variation in the range of about 6%-17%. Hence, the open-aperture Z-scan measurements on 3PA in the perovskite NCs performed at 1050 and 1100 nm further confirm our 3PA results measured with the multi-photon excited upconversion PL technique. The good agreement of the 2PA (as in original SI lines 109-161 and figure S7) and 3PA cross-sections acquired by the multi-photon excited upconversion PL technique with that obtained from open-aperture measurements validates that the multi-photon absorption cross-sections of our perovskite NCs have been properly measured with the multi-photon excited upconversion PL technique.”

New references 30-33 **were added** to the SI:

30. Woodall, M.A. Nonlinear absorption techniques and measurements in semiconductors. *North Texas State University Dissertation*, Ch.4 pg.138 (1985).
31. Brandi, H. S.& de Araujo, C. B. Multiphoton absorption coefficients in solids: a universal curve. *J. Phys. C: Solid State Phys.* **16**, 5929-5936 (1983).
32. He, J., Qu, Y., Li, H., Mi, J. & Ji, W. Three-photon absorption in ZnO and ZnS crystals. *Opt. Express* **13**, 9235-9247, (2005).
33. Sutherland, R. L. with contributions by McLean, D. G. & Kirkpatrick, S. *Handbook of Nonlinear Optics*, New York, NY: *Marcel Dekker*, **Second Edition**, Revised and Expanded, (2003).

Moreover, the related discussion on the open-aperture Z-scan measurements in the original manuscript, lines 109-115 **was revised** to add the discussion on the 3PA measurements on the perovskite NCs

In the original manuscript, lines 109-115: “*The 2PA cross-sections (σ_2 values at 800 nm) of the colloidal NCs (measured using the open-aperture z-scan technique) were employed as a standard for multi-photon excited PL (MEPL) measurements at different wavelengths (see Methods section and SI for more details). Our measured σ_2 values (from z scan) for CsPbBr₃ NCs at 800 nm agrees well with literature reports^{28,29}. The consistency of the ratio of the σ_2 values (for MAPbBr₃, CsPbBr₃ and MAPbBr₃/(OA)₂PbBr₄ NCs at 800 nm) obtained from z-scan (Supplementary **Fig. S4**) with the two-photon-excited PL measurements (**Fig. S6a**), provides further validation of our approach.*”

was revised to:

“The 2PA cross-sections (σ_2 values at 800 nm) of the colloidal NCs (measured using the open-aperture z-scan technique) were employed as a standard for multi-photon excited PL (MEPL) measurements at

different wavelengths (see Methods section and SI for more details). Our measured σ_2 values (from z scan) for CsPbBr₃ NCs at 800 nm agrees well with literature reports^{28,29}. The consistency of the ratio of the σ_2 values (for MAPbBr₃, CsPbBr₃ and MAPbBr₃/(OA)₂PbBr₄ NCs at 800 nm) obtained from z-scan (Supplementary **Fig. S7**) with the two-photon-excited PL measurements (**Fig. S11a**), provides further validation of our approach. Moreover, the good agreement of the measured 3PA cross-sections with those acquired from open-aperture Z-scan measurements at 1050 and 1100nm (see SI for more details) further confirms that the MPA cross-sections have been properly measured with the MEPL technique.” on page 7, paragraph 1, line 2 of the revised manuscript.

In addition, the related discussion on the comparison between multi-photon excited upconversion PL measurement technique and the open-aperture Z-scan technique **was added** after line 171 in the original SI:

“Multi-photon excited upconversion PL measurements have been demonstrated to provide much higher detection sensitivity for characterizing MPA properties of both inorganic semiconductor NCs and organic molecules than open-aperture Z-scan, especially for higher-order MPA and highly luminescent MPA-active materials^{17,34,35}. Due to the low detection efficiency, extremely high excitation peak intensities are required in open-aperture Z-scan for characterizing the high-order multi-photon absorption (*i.e.*, four- and five-photon absorption), which may cause sample damage or introduce other effects (or artifacts) with the extremely high excitation intensities. Consequently, most of the characterizations for the high-order multiphoton absorption (*i.e.*, four- and five-photon absorption) were performed with multi-photon excited upconversion PL measurements³⁵⁻³⁷ and open-aperture Z-scan has been seldom used in this region. Hence, the multi-photon excited upconversion PL measurements have applied to characterize the multi-photon (2-, 3-, 4-, 5-photon) absorption cross-sections of the perovskite NCs.”

New references 34-37 **were added** into the SI:

34. Xu, C., Zipfel, W., Shear, J. B., Williams, R. M. & Webb, W. W. Multiphoton fluorescence excitation: new spectral windows for biological nonlinear microscopy. *Proc. Natl. Acad. Sci. U.S.A.* **93**, 10763-10768 (1996).
35. He, G. S., Tan, L.-S., Zheng, Q. & Prasad, P. N. Multiphoton Absorbing Materials: Molecular Designs, Characterizations, and Applications. *Chem. Rev.* **108**, 1245-1330 (2008).
36. He, G. S. *et al.* Two-, three-, and four-photon-pumped stimulated cavityless lasing properties of ten stilbazolium-dyes solutions. *J. Opt. Soc. Am. B* **22**, 2219-2228, (2005).
37. Fan, H. H., Guo, L., Li, K. F., Wong, M. S. & Cheah, K. W. Exceptionally Strong Multiphoton-Excited Blue Photoluminescence and Lasing from Ladder-Type Oligo(p-phenylene)s. *J. Am. Chem. Soc.* **134**, 7297-7300 (2012).

Comment 5) After check these points and make clear why it is 9 orders of magnitude higher, the authors should send this manuscript for a more specific journal on the photochemical and photophysical field. The giant 5 photon absorption cross-section is not enough information to justify a publication in Nature Communications. The sample is already know and multiphoton absorption as well.

Response 5)

We respectfully disagree with the reviewer on this comment. The novelty of our work are:

1. First report on *giant five-photon absorption and resultant highly efficient upconversion fluorescence* in halide perovskite colloidal NCs excited at IR ($\sim 2 \mu\text{m}$) that overcomes the inherent challenges of small five-photon action cross-sections in organic chromophores and conventional semiconductor NCs.
2. First report on *detailed characterization of the giant multi-photon (2-, 3-, 4-, 5-photon) action cross-sections* of the halide perovskite colloidal NCs over the wide wavelength range of 675 – 2300 nm.
3. First report on further achieving *large enhancement on multi-photon action cross-sections* in halide perovskite colloidal NCs utilizing a *multi-dimensional core-shell structure*.
4. An unprecedented *9-orders larger five-photon action cross-sections* ($\eta\sigma_5 \sim 10^{136} \text{ cm}^{10} \text{ s}^4 / \text{photon}^4$) than *state-of-the-art specially-designed organic molecules* is observed in *multi-dimensional core-shell perovskite NCs*, demonstrating the potential for next generation multi-photon imaging applications with unmatched imaging depth, sensitivity and resolution.

These new insights uncovered in our work challenge the conventional wisdom of multi-photon absorption in traditional colloidal semiconductor NCs and enable fresh approaches for the development of next generation multi-photon imaging applications.

Reviewer's comments to the author

Reviewer 1

The authors have addressed all my concerns in the revised MS. I recommend publishing the manuscript.

Reviewer 2

Following the reviewers' suggestions, the authors provided additional evidence or more experimental details / figures to address the reviews concerns and strengthen their conclusions. The article, in its present form, is much improved. I am very satisfied with the revised version of the manuscript, thus suggest the acceptance of the manuscript without changes.

Reviewer 3

After all questions be well answered by the authors, I decided that this article, in my opinion, can be accepted for publication on Nature Communications.

The three-photon results measured by the authors made the work stronger, show that both techniques, at least for two and three photons are in excellent agreement. All questions were answered accordingly with I was expecting. Also the authors show more results than I had asked, with contributed to solve some doubts that I still had.

Response to all three reviewers

We would like to thank all the reviewers for their valuable time, constructive comments and feedback to help strengthen this work.